



# Northern Hemisphere Contrail Properties Derived from Terra and Aqua MODIS Data for 2006 and 2012

David P. Duda[1], Sarah T. Bedka[1], Patrick Minnis[1], Douglas Spangenberg[1], Konstantin Khlopenkov[1], Thad Chee[1], and William L. Smith, Jr.[2]

[1]Science Systems and Applications, Inc., Hampton, VA 23666-5986, USA
[2]NASA Langley Research Center, Hampton, VA 23681-2199, USA

*Correspondence to*: David P. Duda (david.p.duda@nasa.gov)

**Abstract.** Linear contrail coverage, optical property, and radiative forcing data over the Northern Hemisphere (NH) are derived from a year (2012) of *Terra* and *Aqua* Moderate-resolution Imaging Spectroradiometer (MODIS) imagery, and are

10 compared with previously published 2006 results (Duda et al., 2013; Bedka et al., 2013; Spangenberg et al., 2013) using a consistent retrieval methodology. Differences in the observed *Terra*-minus-*Aqua* screened contrail coverage and patterns in the 2012 annual-mean air traffic estimated with respect to satellite overpass time suggest that most contrails detected by the contrail detection algorithm (CDA) form approximately 2 h before overpass time. The 2012 screened NH contrail coverage (Mask B) shows a relative 3% increase (from 0.136% to 0.140%) compared to 2006 data for *Terra* and increased by almost

15 7% (0.134% to 0.143%) for *Aqua*. A new post-processing algorithm added to the contrail mask processing estimated that the total contrail cirrus coverage visible in the MODIS imagery may be three to four times larger than the linear contrail coverage detected by the CDA. This estimate is similar in magnitude to the spreading factor estimated by Minnis et al. (2013). Contrail property retrievals of the 2012 data indicate that both contrail optical depth and contrail effective diameter decreased approximately 10% between 2006 and 2012. The decreases may be attributed to better background cloudiness

characterization, changes in the waypoint screening, or changes in contrail temperature. The total mean contrail radiative forcing (TCRF) for all 2012 *Terra* observations were -6.3, 14.3, and 8.0 mW m$^{-2}$ for the shortwave (SWCRF), longwave (LWCRF), and net forcings, respectively. These values are approximately 20% less than the corresponding 2006 *Terra* estimates. The decline in TCRF results from the decrease in normalized CRF, partially offset by the 3% increase in overall contrail coverage in 2012. The TCRFs for 2012 *Aqua* are similar, -6.4, 15.5, and 9.0 mW m$^{-2}$ for shortwave, longwave, and

net radiative forcing. The strong correlation between the relative changes in both total SWCRF and LWCRF between 2006 and 2012 and the corresponding relative changes in screened contrail coverage over each air traffic region suggests that regional changes in TCRF from year to year are dominated by interannual changes in contrail coverage over each area.

## 1 Introduction

Persistent linear contrails are aircraft-generated clouds that can form in ice-supersaturated zones of the upper

troposphere and add to the naturally occurring cirrus coverage in air traffic regions. As air traffic has increased, studies (*e.g.,*



Minnis et al., 2004; Eleftheratos et al., 2016) have observed increases in cirrus coverage in air traffic corridors, prompting research into the possible impacts of aviation on climate. Several studies have used satellite remote sensing to quantify linear contrail coverage (CC) and to determine the optical properties of these clouds. Mannstein et al. (1999) developed an automated contrail detection algorithm (CDA) to detect linear contrails from Advanced Very High Resolution Radiometer

(AVHRR) imagery and estimated CC over Western Europe. Meyer et al. (2002) also derived CC from AVHRR data over the same region, and later over Southeast Asia (Meyer et al., 2007). Minnis et al. (2005) used the Mannstein et al. CDA and remote sensing methods to estimate contrail optical properties including optical depth ($\tau$) and effective particle diameter ($D_e$) over the north Pacific from AVHRR data. As part of ACCRI (Aviation Climate Change Research Initiative – Brasseur et al., 2016), a consistent analysis system was developed by using a single set of satellite data and algorithms to detect linear

contrails (Duda et al., 2013), to retrieve their particle sizes and optical depths (Bedka et al., 2013), and to produce a concurrent analysis of the atmospheric and surface conditions that enable calculation of contrail radiative effects (Spangenberg et al., 2013) for the detected linear contrails.

Duda et al. (2013) developed a modified version of the Mannstein et al. CDA with three levels of sensitivity (Masks A, B, and C) to investigate how to distinguish linear contrails from natural background linear features. Based on subjective

visual analyses by a team of observers, the mid-range Mask B was found to have the best overall balance between falsely detected and missed contrails. The contrail masks were applied to a year (2006) of 1-km *Aqua* Moderate-resolution Imaging Spectroradiometer (MODIS) imagery taken over the Northern Hemisphere (NH). To improve the detection accuracy, particularly over more difficult regions such as the Tropics where tropical cirrus streaks are common, the contrail mask results were screened using actual commercial flight route information to eliminate false detections in areas lacking air

traffic. These results represented the first hemispherical analysis of linear CC from observations. The greatest coverage was found over the North Atlantic rather than the heaviest air traffic regions (CONUS [contiguous United States] and Europe) due to several factors, including the difficulty of detecting linear contrails when they overlap in high density air traffic regions, the increased sensitivity of the CDA over the ocean due to the relative homogeneity of the oceanic background in the thermal IR (Mannstein et al., 1999; Meyer et al., 2002), and the improved contrail detection in this region due to the

broad, parallel spacing of North Atlantic flights. The NH coverage from Mask B (0.135%) was found to be less than that estimated in previous studies. Bedka et al. (2013) determined the contrail $\tau$ and ice particle $D_e$ from the MODIS data assuming that the contrails were composed of distributions of hexagonal ice columns, while contrail temperatures were estimated based on mean flight levels. The mean NH contrail $\tau$ and $D_e$ and were found to be 0.216 and 35.7 µm, respectively. Both results are similar to other studies (*e.g.,* Iwabuchi et al., 2012, Minnis et al., 2005). Both $\tau$ and $D_e$ tended to decrease

with decreasing contrail temperature.

The satellite-retrieved contrail properties were then used by Spangenberg et al. (2013) to compute contrail radiative effects for the detected linear contrails. Spangenberg et al. performed a cloud analysis of the same MODIS data to provide concurrent background cloud and surface parameters for contrail radiative forcing (CRF) calculations using the CC mask and optical property data. The CC and optical properties were used along with the cloud property data to compute the CRF for 4



months of *Aqua* MODIS data during 2006. The greatest net CRF occurred at night because the longwave and shortwave forcings tend to cancel each other during the day. The greatest regional forcing was found over the North Atlantic and Persian Gulf. Overlapping contrails diminished the linear contrail net CRF over areas with heavier air traffic due to missed linear contrail detections. Overall, the Northern Hemisphere net CRF is 10.6 mW m$^{-2}$ from the Mask B results, which is

smaller than most climate model estimates.

Although the ACCRI studies advanced knowledge of contrail radiative forcing and the impact of contrails on climate, several questions remained because of the limited temporal range of the study. More specifically, the original study could not investigate interannual changes in CC. Air traffic has increased by roughly 5% per year for many years (Lee et al., 2010), and contrail cirrus coverage is likely to have also increased. In addition, it is unclear how much interannual changes

in upper tropospheric meteorological conditions may affect the global CC. Finally, the magnitude of non-linear contrail cirrus coverage, which cannot be detected by the CDA, is still poorly understood and has usually been studied regionally (Minnis et al., 2013; Schumann and Graf, 2013).

To help address these uncertainties, we have derived a second year (2012) of linear contrail property and radiative forcing data over the NH from *Terra* and *Aqua* MODIS imagery. This research uses the same contrail detection

methodology from the ACCRI study to extend our climatology of linear contrail properties, and examines how CC, optical properties, and radiative forcing have changed when compared to the 2006 data. A separate post-processing algorithm was also developed to estimate contrail cirrus coverage by detecting contrail-like cirrus clouds in the vicinity of the linear contrails found by the CDA, which allows us to evaluate better the impact of non-linear contrail cirrus on climate.

## 2 Methodology

### 2.1 Contrail mask

The CDA employed in this study is nearly the same as the algorithm used in Duda et al. (2013). A modified form of the Mannstein et al. (1999) method, it uses data from five (6.8, 8.5, 11.0, 12.0, and 13.3 μm) thermal infrared MODIS channels to reduce the occurrence of false positive detections. Both MODIS instruments are nearly identical, and the NASA MODIS Characterization Support Team is responsible for maintaining the calibration of both satellite sensors to the same

standard (Xiong et al., 2015). As a result, the differences in the calibrated thermal IR brightness temperatures between both satellites are small enough that the same CDA can be used on both *Terra* and *Aqua* imagery. Global aircraft emissions waypoint data provided by the FAA (similar to the data provided in Duda et al. (Wilkerson et al., 2010)) allow comparison of detected contrails with commercial aircraft flight tracks to screen out false detections. The waypoint data and U-V wind component profiles from Modern Era Retrospective Analysis for Research and Applications (MERRA, see Rienecker et al.,

2011) reanalyses are combined to produce a pixel level product of advected flight tracks that are used to assign a confidence of contrail detection for the contrail mask. In addition, a new post-processing method was applied to detect non-linear





contrail cirrus missed by the CDA by assuming that cirrus pixels adjacent and with similar radiative signatures to the detected linear contrails were also formed by aircraft emissions.

Between the 2006 and 2012 analyses, only a few minor changes were made to the linear contrail mask. Two unavoidable alterations to ancillary data sources were required. Because the Goddard Earth Observing System Data

Assimilation System (Bloom et al., 2005) data stream ended in December 2007, MERRA reanalysis data were used to advect the contrail flight tracks. In addition, some changes were reported in the commercial aircraft waypoint data used to create the flight tracks. For the 2006 waypoint data, in regions of the NH where only flight plan data were available (generally, all areas outside of CONUS and the western North Atlantic), great circle routes were used with a random dispersion added so that the flights had a little variance. The 2012 waypoint data used the shortest-distance flight route without any variation for

all non-radar flights so that most oceanic flights during each day were along virtually identical flight routes. Mask B from the original analysis is used in this study as the standard mask to determine interannual variations in linear contrails, because it provides the least biased contrail detection frequency based on a subjective visual analysis of contrails in a randomly selected set of 40 granules (5-minute segments of MODIS imagery) by a team of human observers (Duda et al., 2013).

Some modifications to the contrail mask were also introduced in the processing of the 2012 data to allow for the use

of the contrail cirrus post-processing algorithm. To estimate contrail cirrus coverage in the 2012 data, two new masks (labeled Mask D and E) were developed that use a two-step method to evaluate contrail cirrus coverage. The principle behind the new masks is to first detect linear contrails using a conservative mask, and then to find pixels with similar brightness temperature (BT) or brightness temperature difference (BTD) values within an arbitrary pixel distance (currently 105 pixels) from each detected linear contrail pixel, identifying these similar pixels as contrail cirrus pixels. Masks D and E

were tuned to minimize respectively the false detection of surface features over desert and the false detection of tropical cirrus streamers as contrails. The post-processing algorithm was applied only to the conservative Masks D and E to minimize the possibility of adding non-contrail cirrus to the contrail cirrus estimates. For each candidate pixel in a satellite image, the algorithm uses the mean BT/BTD properties of nearby detected linear contrail pixels (currently within 50 pixels) to serve as the standard to determine which prospective pixels are similar enough to be considered contrail cirrus. The new

contrail cirrus detection algorithm contains several adjustable parameters to vary the size of the area to search for possible contrail cirrus, to vary the size of the area to look for nearby detected linear contrail pixels to serve as a comparison standard, and to change the magnitude of each BT threshold. The current version of the algorithm uses seven BT/BTD thresholds for the following combinations of thermal IR images: $13.3^{i}$ μm; $11 - 12$ μm; $8.6 - 12$ μm; $12^{i}$ μm; $(8.6 - 13.3$ μm$) + (11 - 12$ μm$) - 12$ μm; $(11 - 12$ μm$) - 13.3$ μm; $6.8^{i}$ μm, where the $i$ superscript denotes that the brightness temperature in the channel

is inverted (*i.e.,* the negative of the temperature is used so that both BT and BTD images can be treated with the same procedures). Several sensitivity tests were run on a sample of 24 *Aqua* MODIS granules from 2012 and 8 *Terra* MODIS granules from 2006 covering several scene types including desert, Arctic, tropical ocean and high-air traffic regions over the North America, the North Atlantic, Europe and China. The best set of adjustable parameters was found through a process of





trial and error, by visually inspecting each combination of parameters that was run, and determining the combination with the best overall results for all granules.

**2.2 Contrail property retrievals**

To determine interannual differences in contrail properties, the contrail analysis system should ideally remain as consistent and unchanging as possible. Due to the complex analysis system used to detect linear contrails and then to analyze the NH contrail properties, some changes were unavoidable. In addition to the changes in the ancillary data sources noted above, some updates and minor improvements of the background cloud characterization software were implemented. While the contrail property retrieval algorithm did not change between this study and the earlier study using the 2006 data (Bedka et al. 2013), improvements were made to the determination of background cloud properties, most notably improving

the accuracy of cloud phase. These changes produced a reduction in the percentage of pixels with "indeterminate background" from several percent to nearly zero. The percentage of ice cloud backgrounds also increased, bringing the results closer to the ice cloud coverage estimated by the rigorously validated CERES cloud algorithm (CERES, 2016; Trepte et al., 2018). Several other changes were made to address software errors that had been discovered.

**2.3 Contrail radiative forcing**

The contrail radiative forcing calculations used the same procedure as with the 2006 data study (Spangenberg et al., 2013). An updated version of the Fu-Liou radiative transfer program was employed, but it is not expected to affect the computed radiative forcing significantly. The most important change to the CRF assessment would be due to differences in the determination of the background cloud properties discussed above. No CRF calculations were possible for the post-processed contrail cirrus Masks D and E due to difficulties in determining cloudiness background in situations where contrail

cirrus extended over a large area.

**3 Results**

**3.1 Contrail mask**

The first and most basic parameter determined from this study is CC. The CC mask determines the amount and location of linear contrails and provides the foundation for the subsequent contrail property and radiative forcing retrievals.

The results of CC Mask B for 2012 are summarized in Figure 1.

Some consistent differences appear between the *Terra* (with overpasses at approximately 10:30 and 22:30 local time) and *Aqua* (with overpasses at approximately 01:30 and 13:30 local time) coverage in 2012. For example, *Terra* coverage is greater than *Aqua* over most air traffic regions including CONUS, Europe, China, and the eastern half of the air route between Hawaii and western CONUS. *Aqua* coverage is greater than *Terra* over the central North Atlantic, portions of

the Europe to Latin America (LA) air route, northern Asia, and northern Africa.



Assuming that on average the upper tropospheric temperature and humidity do not change significantly during the 3 hours between the two overpass times, then the differences found in the *Terra* and *Aqua* screened CC are mostly likely due to differences in air traffic density. Figure 2 shows the *Terra* minus *Aqua* annual mean air traffic density difference estimated at 1, 2, 3, and 4-h before the overpass times. The best match between the patterns in Figure 2 and Figure 1 appears

to vary from 1 to 3 h, depending on location. Overall, good matches occur for the case where the 2012 annual-mean air traffic densities are 2 h before the overpass times, suggesting that most contrails are about 2 h old when detected by the satellite CDA. Previous studies (Duda et al., 2004, Vázquez-Navarro et al., 2015) however, have reported a contrail mean age of 1 h in contrails identified in geostationary satellite data, indicating that many mid-latitude contrails are detectable as early as 1 h after formation.

By comparing the results of this study with the 2006 data, we can examine interannual changes in linear CC. The 2012 screened *Terra* Northern Hemisphere CC (Mask B) shows a 3% relative increase compared to 2006 data, from 0.136 percent to 0.140 percent, while the 2012 screened *Aqua* coverage increased by almost 7 percent, from 0.134 percent to 0.143 percent.

An examination of the changes in screened annual mean NH CC between 2006 and 2012 (Figure 3) versus the

15 changes in unscreened CC (Figure 4) provides some insight into how air traffic density affected the screened coverage estimations. In comparison with the screened CC, the hemispheric-mean unscreened CC changed only slightly between 2006 and 2012. *Terra* NH CC decreased 2 percent compared to 2006 data from 0.337% to 0.329%, and the 2012 unscreened *Aqua* CC increased 1 percent, from 0.312% to 0.316%. Both satellites show similar changes in the magnitude and distribution of the screened and unscreened CC between 2006 and 2012. Both the screened and unscreened CC in Figures 3 and 4 show

larger increases along the North Atlantic corridor and parts of the Indian Ocean, with smaller increases over northwestern CONUS, northwestern Asia, and tropical Africa. Decreases in screened and unscreened CC from 2006 to 2012 are apparent over southern CONUS, Western Europe, and northeastern Canada. The most notable differences between Figures 3 and 4 occur in the air routes between Europe and Latin America, and between CONUS and Hawaii, where screened CC decreases between 2006 and 2012, while the unscreened CC changes are mixed.

To examine the interannual differences in CC more closely, the NH was divided into nine air traffic regions to determine where CC changes were most pronounced (Figure 5). Table 1 summarizes the relative changes ([2012 – 2006]/2006) in the screened and unscreened CC (day+night) of the Northern Hemisphere for each of the nine regions.

The most prominent differences between the screened and unscreened interannual CC are in the transoceanic air traffic regions (HI/CONUS, Europe/LA) and western Asia. These two transoceanic regions have significant declines in

screened coverage while the corresponding unscreened coverage changes are smaller. In contrast, western Asia shows moderate declines in unscreened coverage but small increases in screened coverage. To investigate whether the differences between the screened and unscreened CC trends in these air traffic regions can be explained by changes in the air traffic density between 2006 and 2012, Figure 6 shows the interannual difference in annual-mean air traffic density between 2012 and 2006 at 2 h before *Terra* and *Aqua* overpass time. As presented above, an examination of the differences in the *Terra*





and *Aqua* screened CC for 2012 suggests that most detected contrails form approximately 2 h before satellite overpass time. Figure 6 shows that air traffic density increased over nearly all of the Northern Hemisphere between 2006 and 2012 except for the air corridor between Europe and Latin America, and over parts of the HI/CONUS corridor.

Table 2 provides a list of the mean air traffic changes between 2006 and 2012 based on a sample of waypoint data from the nine air traffic regions. Seven of the nine air traffic regions show increases in air traffic. The largest increases occur over E Asia and W Asia. Two regions show a decrease in air traffic: a small decrease in the Hawaii to CONUS corridor, and a nearly 60 percent decrease in the Europe to LA corridor. Thus, the decrease in air traffic in these two regions results in diminished screened CC but smaller changes in unscreened CC. [The large increase in air traffic over western Asia may also explain in part why the screened coverage remained relatively unchanged in 2012 despite the observed moderate decrease in
unscreened coverage.]

Like changes in air traffic, interannual changes in the upper tropospheric thermodynamic state also affect the detected linear CC. To consider this possible factor, we present in Figure 7 and Table 2 the interannual change in potential persistent contrail frequency (PPCF) between 200 and 250 hPa, the tropospheric layer where most of the contrail-forming air traffic occurs. The PPCF is computed using temperature and relative humidity statistics from ERA-Interim (ECMWF)
reanalysis data (Dee et al., 2011), and is an indicator of how often conditions that are favorable for the development of persistent contrails occur. It is assumed that MERRA relative humidities are consistent with their ERA-Interim counterparts.

The relative differences in PPCF between 2006 and 2012 are listed in Table 2. In many of the air traffic regions, the relative differences in PPCF between 2006 and 2012 are small, suggesting that the differences between the 2006 and 2012 screened CC in those regions might be only slightly affected by the interannual changes in the thermodynamic state of the
upper troposphere. However, for some regions, the interannual changes in PPCF are more significant, and may have played a larger role in the difference in number of detected contrails between 2006 and 2012. The PPCF changes (Figure 7) correlate with unscreened CC changes (Figure 4) over northwestern and central Asia, the Indian Ocean, southern CONUS, northern Europe and off the coast of Western Europe, Greenland, and parts of northern Canada. For the air traffic regions, the increase in screened and unscreened coverage over the North Atlantic is correlated with an increase in PPCF between
2006 and 2012, but in the Europe to LA air corridor the increase in PPCF in 2012 appears to have minimal impact on the decrease in screened CC.

The results presented here demonstrate that interannual changes in air traffic density and PPCF appear to have some influence on the change in screened satellite-detected CC between 2006 and 2012, although some of the interannual changes are more difficult to explain, especially the large increase in screened and unscreened coverage over the North Atlantic air
corridor. The increase in CC between 2006 and 2012 over the North Atlantic may be due to changes in flight altitudes in 2012 that shifted more flights into levels of the atmosphere where ambient conditions are more likely for persistent contrail formation than in 2006.



### 3.2 Contrail cirrus coverage estimation

In addition to linear contrails, CC due to contrail cirrus would increase the overall contrail radiative forcing (CRF) as contrails spread into non-linear, overlapping cloudiness that cannot be detected by the CDA. To estimate contrail cirrus effects, Minnis et al. (2013) tracked contrails over the United States in geostationary satellite imagery, and determined the
properties of contrail cirrus over selected areas; it was determined from visual inspection that only contrails produced the existing cirrus clouds in each region. Overall for 21 cases, the combined linear and contrail cirrus coverage was on average 3.5 times the value determined from mask B. The contrail cirrus $\tau$ and $D_e$ values were larger than the corresponding linear contrail values.

An example of the two new masks D and E that were developed to estimate contrail cirrus coverage is presented in
Figure 8 for a 5-minute *Terra* MODIS granule starting at 1210 UTC on 18 January 2012. Figure 8a shows the $11 - 12$ µm BTD image, which highlights a group of linear contrails off the coast of Western Europe. Figure 8b presents the screened coverage results from Mask B, including the detected contrails off the coast, and also contrails over Great Britain, Ireland and portions of France and the Iberian peninsula. Figures 8c and 8d display the results from Masks D and E respectively, with the linear contrails and estimated contrail cirrus shown in different colors for clarity. Both masks detect areas of diffuse
cloudiness near the identified linear contrails. The optical properties of contrails approach those of natural cirrus clouds as they age (Mannstein and Schumann, 2005), thus Masks D and E can only measure nearby contrail-like cirrus, but they might also include natural cirrus. However, both masks provide a method to estimate the amount of cirrus coverage associated with linear contrails, which is nevertheless useful for observing how contrail-associated cirrus varies between different regions and satellite observation times.

Figure 9 shows the ratio between the screened linear CC from Masks D and E and the total contrail cirrus coverage computed by the algorithm. The ratio provides an estimate of how much additional cirrus coverage may be attributed to contrail cirrus formation, and is referred to as the spreading factor (*SF*) in Minnis et al. (2013). As mentioned earlier, Masks D and E were developed to minimize the false detection of surface features and tropical cirrus streaks, respectively. Mask E is especially conservative to minimize the effects of tropical cirrus, and thus represents a lower bound estimate of contrail
cirrus, while Mask D represents an upper bound estimate of contrail cirrus coverage. In high air traffic regions, the coverage ratio is consistently larger for *Aqua* data compared to corresponding *Terra* data for both masks, especially during the day. At least part of the daytime difference in *SF* between *Terra* and *Aqua* is due to the greater air traffic over CONUS and Europe 2 h before the *Aqua* satellite overpass time when most contrails that are visible to the satellite form.

The NH-mean contrail cirrus coverage ratios during the daytime and nighttime (not shown) are similar. For *Terra*
Mask D the hemispheric mean *SF* during the day (3.80) and during the night (3.82) are nearly identical, while for *Aqua* Mask D the daytime NH-mean *SF* is 4.38 and 4.30 at night. The NH-mean estimates of *SF* for all times (*i.e.,* day+night) and both satellites range from 4.34 to 3.80 for Mask D, and from 3.53 to 3.11 for Mask E. These estimates are roughly comparable to the mean *SF* (3.5) determined by Minnis et al. (2013).



In tropical ocean regions with significant CC [Arabian Sea, Bay of Bengal NW of Sri Lanka, South China Sea], the *SF* exceeds that of the neighboring land areas. The land/ocean discrepancy is especially visible in the *Aqua* observations and is larger during the day. The land/ocean discrepancy is not apparent at higher latitudes in either satellite, and is probably the result of the BT/BTD thresholds masking out more tropical cirrus streaks over land than over ocean.

5 **3.3 Contrail radiative property retrieval results**

The seasonal and annual mean contrail properties retrieved from *Terra* and *Aqua* MODIS data for 2006 and 2012 Mask B screened CC are summarized in Table 3. The contrail $\tau$ retrieved from both *Terra* and *Aqua* data decreased from 2006 to 2012, both during the day (*Terra*: 14%; *Aqua*: 15%) and during the night (*Terra*: 9.3%; *Aqua*: 11%). The overall probability distribution of contrail $\tau$ was nearly the same for both years, with most contrails having $\tau$ less than 0.2 (Figure 10 10). Most notable in the 2012 data was the decrease in the number of large $\tau$ contrails (contrails with $\tau > 0.5$), which covered 7.5% of the 2006 dataset and only 6.25% of the 2012 dataset. The decrease might be attributed to better background characterization and/or differences resulting from the changes in waypoint screening. According to the waypoint data for 2012, the NH-mean contrail temperature decreased 1.6 K compared to 2006 due to an overall mean increase of 0.26 km in flight track altitudes. Table 2 also reports regional changes in contrail temperature. An increase in contrail height could 15 result in thinner contrails because of the overall decrease in contrail temperature.

The 2012 *Terra* mean $D_e$ decreased 10.6% (3.6 μm) during the day and 9.6% (3.3 μm) during the night compared to the 2006 results, while for *Aqua* the 2012 mean daytime $D_e$ decreased 8.4% (3.0 μm) and the 2012 mean nighttime $D_e$ decreased 7.6% (2.7 μm) relative to the corresponding 2006 values. The number of contrails with $D_e < 24$ μm increased substantially in the 2012 data (Figure 11). The largest increase was in the 8 – 16 μm bin, where the percentage of contrails 20 increased from 11.5% in 2006 to nearly 17% in 2012 for the *Terra* data. The number of *Terra* contrails with very large $D_e$ (>80 μm) decreased from 7.5% in 2006 to 5% in 2012, which may be attributed once again to either better background characterization or differences in flight track screening. The NH map of the difference in mean contrail $D_e$ between 2012 and 2006 (not shown) shows that the 2012 particle sizes are systematically smaller across the entire hemisphere with the largest decreases in the Arctic, confirming that the differences in retrieved $D_e$ between 2006 and 2012 likely result at least in part 25 from changes in background characterization.

Bedka et al. (2013) indicated that a 1 K decrease in contrail temperature led to a 5.6% decrease in retrieved $\tau$ and a 1% increase in $D_e$. To investigate whether improvements and bug fixes to the background cloud property determination have affected the $\tau$ and $D_e$ retrievals, two months of 2006 *Aqua* MODIS data (Jan and Jul) were reprocessed and compared with the original 2006 results. The two months of reprocessed *Aqua* retrievals from 2006 suggest that the updates to the 30 contrail property retrieval code could be responsible for up to 3% of the reduction in $\tau$, and 6% of the reduction in $D_e$. These differences, coupled with the observed differences in contrail temperature, may explain most of the differences in $D_e$ and $\tau$ between 2006 and 2012.





### 3.4 Contrail radiative forcing results

The annual mean NH radiative forcing due to linear contrails (CRF) was computed for 2012 *Terra* and *Aqua*
MODIS data, and is presented in Table 4 along with the 2006 results. The 12-month mean normalized (that is, the CRF
assuming a CC of 100%) shortwave, longwave, and net radiative forcings for *Terra* 2012 are -4.7, 10.8, and 6.1 W m$^{-2}$. The

2012 normalized CRFs are approximately 20 percent less than the normalized CRFs computed from the 2006 *Terra* data.
The decrease in normalized forcing is due in part to the decrease in mean contrail τ (Meerkotter et al., 1999) in the 2012 data.
Sensitivity tests computed by Spangenberg et al. (2013) suggest that decreases in $D_e$ would also slightly diminish the
longwave forcing but increase the shortwave forcing marginally, while the increase in mean contrail altitude would partly
compensate for the reduced τ. The normalized CRFs for *Aqua* are similar. The shortwave, longwave, and net radiative

forcings for *Aqua* 2012 are -4.7, 11.5, and 6.8 W m$^{-2}$, about 20% smaller than the corresponding CRFs for *Aqua* 2006.

Changes in background scene type may also have contributed to the reduction in normalized CRF. An increase in
the percentage of contrails over ice clouds (especially at night) between 2006 and 2012 (day+night: from 34.2% to 46.4%
*Terra*; 31.6% to 38.9% *Aqua*) and a decrease in the fraction of clear-sky contrails (day+night: from 19.0% to 15.2% for
*Terra* and 20.0% to 16.7% for *Aqua*) may have also reduced the normalized CRF because radiative forcing tends to be

greater in clear skies compared to cloudy skies (Spangenberg et al., 2013).

The total contrail radiative forcing (TCRF) is computed by multiplying the normalized CRF by the contrail fraction
for each image and summing for all images, thus providing a realistic estimate of contrail radiative forcing at the time of
*Terra* overpasses. The 12-month mean (day+night) total shortwave, longwave, and net radiative forcings are -6.3, 14.3, and
8.0 mW m$^{-2}$ for the 2012 *Terra* data, which are 12 to 24% less than the corresponding 2006 *Terra* values. The smaller TCRF

for 2012 results from the combination of an approximately 20% decrease in normalized CRF partially offset by a 3%
increase in overall CC. For *Aqua*, the annual-mean (day+night) total shortwave, longwave, and net radiative forcings are -
6.4, 15.5, and 9.0 mW m$^{-2}$ for the 2012 data, which are 10 to 16% less in magnitude than the corresponding 2006 *Aqua*
values. The decrease in the 2012 *Aqua* TCRF results from the approximately 20% decrease in normalized CRF, partially
compensated by a 7% increase in overall CC.

Table 5 presents the relative changes in total SWCRF, LWCRF, and net CRF (day+night) for the Northern
Hemisphere and each of the nine air traffic regions. For both *Terra* and *Aqua*, SWCRF decreased roughly 10% over the NH.
The magnitude of the SWCRF also shrank in each of the nine air traffic regions, except for a small increase in the North
Atlantic for *Terra* due to the nearly 22% increase in CC in 2012. The largest decrease in SWCRF occurred in the Europe to
Latin America air route where the largest drop in CC was also reported. Larger than average declines are apparent over

Europe and in the HI-to-CONUS air corridor, caused at least partially by the larger reductions in 2012 CC in those regions.
Similar changes are evident in LWCRF. LWCRF decreases in 2012 compared to 2006 in all nine air traffic regions in the
NH, with the largest differences occurring in the Europe to Latin America air corridor, and larger than average losses over
Europe and the HI-to-CONUS air corridor. The 2012 versus 2006 differences in net CRF are similar to the changes in



LWCRF because LWCRF tends to be larger than the SWCRF and thus dominates the interannual change in net CRF. The decreases in net CRF in 2012 are generally larger for *Terra* compared to *Aqua* except for the NW Pacific where the *Aqua* difference is slightly larger.

To demonstrate the impact of the regional changes in the screened CC between 2006 and 2012 on the regional changes in the total contrail radiative forcing, Figure 12 plots the relative change in total SWCRF and LWCRF for *Aqua* (*Terra* differences are similar) between 2006 and 2012 as a function of the relative change in screened CC for the NH and the nine air traffic regions shown in Table 5. The relative changes in both SWCRF and LWCRF between 2006 and 2012 correlate well with relative changes in screened CC over each air traffic region. The relatively high correlations suggest that the regional changes in total contrail radiative forcing from year to year are dominated by interannual changes in CC over each area. Other factors that could change contrail radiative forcing (contrail $\tau$, contrail $D_e$, or changes in the background cloud conditions) changed more uniformly across the NH between 2006 and 2012 when compared to CC.

## 4 Discussion

This study analyzes MODIS thermal IR imagery from 2006 and 2012 to investigate interannual changes in contrail properties over the Northern Hemisphere. The 2012 retrievals provide us some insight into the relative impact of air traffic, meteorological, and flight track screening changes on the detection of linear contrails from satellite imagery. The 2012 data show large changes in air traffic distribution across the Atlantic Ocean, including a 60% decrease in air traffic between Europe and Latin America compared to 2006 that is not reflected in the unscreened CC. Although the differences between the screened and unscreened coverage in the Europe/LA corridor suggest that flights were likely missed in the 2012 air traffic inventory, a check of multiple flight schedule data sources for the year (A. Malwitz, personal communication) showed no irregularities that would indicate dropping of flights for the Europe to Latin America region. Therefore, the overall reliability of the air traffic is assumed here to be similar to the dependability of the 2006 waypoint data. The mean aircraft cruise altitude on average rose by 0.26 km over the NH, with increases as large as 0.79 km over the NW Pacific region. Fichter et al. (2005) estimated that lifting flight levels globally by 2 kft (0.61 km) would tend to increase global CC by 6%, with most of the increase at middle and low latitudes and a decrease at high latitudes. Remarkably, the regional changes in screened CC between 2006 and 2012 match best the scenario in Fichter et al. where flights are displaced 2 kft *downward* rather than upward as suggested by the 2012 waypoint data. A comparison of the 2006 versus 2012 ECMWF PPCF changes compared to the PPCF differences computed from MERRA reanalyses (not shown) show little agreement over the North Atlantic, suggesting that details of the upper tropospheric temperature and humidity are uncertain in this region.

A comparison of the changing air traffic density patterns between the *Terra* and *Aqua* overpass times suggests that the mean age of contrails visible in MODIS imagery appears to be approximately 2 h. Thus, the CDA likely can only detect a subset of the larger, more long-lived linear contrails, which impacts the contrail properties and radiative forcing estimated in



this study. Actual contrail numbers are likely larger in high air traffic regions with intersecting flights that make automated detection difficult.

Two competing effects are expected to influence contrail optical depths in the 2012 results. Flight altitudes in 2012 were generally higher than in 2006, so we would expect the contrails in 2012 were forming in colder environments with less
available water vapor, thus with smaller optical depths. In 2006, significantly fewer contrails were detected over ice clouds and more contrails occurred in clear skies. Because contrails over ice clouds in both studies have the highest optical depths, and contrails in clear skies have the lowest, the impact of background cloudiness changes on contrail $\tau$ would be to counteract the effect of higher flight altitudes.

While no calculations of contrail cirrus radiative forcing were made, the total radiative forcing by contrail cirrus is
10 expected to be proportional to the spreading factor, about 3.5 times the total CRF estimated from the detected linear contrails. For current study, a NH total net CRF of 8.5 mW m$^{-2}$ would translate to a global estimate of 4.6 mW m$^{-2}$ assuming that 93% of high altitude air traffic is within the NH (Duda et al. 2013). Thus, the global total net CRF for contrail cirrus would be about 16 mW m$^{-2}$ if contrail cirrus has the same properties as linear contrails. Minnis et al. (2013) found that the mean optical depths of contrail cirrus were 2–3 times greater (and $D_e$ 20% greater) than for adjacent linear contrails, thus
contrail cirrus would have a global total net CRF on the order of 40 mW m$^{-2}$ due to the larger optical depths in contrail cirrus. Recent estimates of global total net CRF for contrail cirrus range from 50-60 mW m$^{-2}$ [Lee et al. 2010].

**5 Future Work**

Although the creation of a two-year NH contrail climatology has allowed us to determine the radiative impact of contrails more completely, further research is needed. Although analyzing more years of data is desirable, the analysis was
20 laborious such that only one year of new results was possible for this study. Regardless, we believe the analysis methods and results are unique and could be employed in future studies, including additional data if the resources were available. Neither 2006 nor 2012 had a major El Niño or La Niña event, while in 2015 a significant El Niño developed in the Pacific Ocean, producing large-scale changes in atmospheric humidity throughout the Tropics and into the mid-latitudes. In particular, strong El Niños are usually associated with a more active sub-tropical jet over southern CONUS, increasing the probability
of cirrus (and persistent contrails) in this region. From ECMWF analyses (not shown) the annual-mean PPCF over southern CONUS is 2–4% higher in 2015 compared to 2012, with even larger regional differences over northern Asia. A future study during this time period would extend the contrail climatology to three years, and would improve our understanding of how interannual meteorological variability may affect contrail cirrus formation.

The algorithm developed to detect contrail-like cirrus in this study is a preliminary attempt to define contrail cirrus.
Although useful as a heuristic tool to examine how contrail cirrus detection varies between different times and locations, it requires refinement, especially over different surface backgrounds and varying viewing angles. Furthermore, the study of contrail cirrus development could be aided by the launch of next generation imagers onboard the Himawari-8 and GOES-16



satellites. These platforms can provide full disk 10 and 15-minute loops, respectively, of high-resolution multi-spectral geosynchronous imagery that would allow detailed analysis of contrail spreading by identifying individual contrails with specific flights from the waypoint database. Expanding on the analysis presented in Minnis et al. (2013), the lifecycles of a large set of identified contrails could be related to several meteorological variables (such as RHI, vertical wind, wind shear,

depth of the super-saturated layer) to determine which factors influence the growth and spreading of persistent linear contrails into contrail cirrus. Such a dataset would help us improve our understanding of how contrail cirrus contributes to the observed increase in global cirrus coverage, and would provide valuable data to contrail models that explicitly simulate the lifespan of contrails, thus advancing our knowledge of the impacts of contrail cirrus on climate.

**Author contribution**

David Duda prepared the manuscript with contributions from co-authors. David Duda, Sarah Bedka, and Doug Spangenberg performed the analysis of the contrail masks, contrail property retrievals, and contrail radiative forcing, respectively. Patrick Minnis conceptualized the overall research goals and aims of the study, supervised the research project and acquired funding for the study. Konstantin Khlopenkov developed programming code for analysing the commercial aircraft waypoint data and advecting contrail tracks via the MERRA wind data. Thad Chee managed the implementation of

the computer code and supporting algorithms to process the satellite data and to create the contrail masks. William Smith also supervised the research project and provided review and commentary of the manuscript.

**Competing interests**

The authors declare that they have no conflict of interest.

**Acknowledgements**

The waypoint data used for this work were provided by U.S. DOT Volpe Center and are based on data provided by the U.S. FAA and EUROCONTROL in support of the objectives of the International Civil Aviation Organization (ICAO) Committee on Aviation Environmental Protection CO2 Task Group. Any opinions, findings, and conclusions or recommendations expressed in this material are those of the authors and do not necessarily reflect the views of the U.S. DOT Volpe Center, the U.S. FAA, EUROCONTROL, or ICAO.

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



**Table 1: Relative change in CC between 2012 and 2006 for the Northern Hemisphere and nine air traffic regions.**

| Region | *Aqua* screened | *Terra* screened | *Aqua* unscreened | *Terra* unscreened |
|---|---|---|---|---|
| NH | +6.92 % | +3.19 % | +1.15 % | -2.39 % |
| N Atlantic | +24.2 % | +21.9 % | +23.6 % | +23.9 % |
| CONUS | -5.02 % | -6.22 % | -1.93 % | -4.63 % |
| Europe | -10.4 % | -8.98 % | -11.8 % | -11.2 % |
| W Asia | +1.64 % | +0.63 % | -9.05 % | -9.66 % |
| E Asia | +5.46 % | +0.34 % | +1.91 % | -1.67 % |
| Europe/LA | -41.0 % | -51.2 % | +9.58 % | -1.54 % |
| HI/CONUS | -18.4 % | -18.3 % | -0.74 % | -6.24 % |
| NE Pacific | +4.08 % | -9.56 % | +3.89 % | +5.56 % |
| NW Pacific | +3.34 % | +6.11 % | -1.30 % | +3.92 % |

**Table 2: Interannual changes (2012 – 2006) in annual-mean flight altitude (in km), annual-mean flight temperature (in K), and the ratio of annual-mean air traffic density between 2012 and 2006 for NH and nine air traffic regions based on a 12-day sample (15th of each month) of waypoint data from both years. Relative interannual changes (%) in annual-mean potential persistent contrail frequency (PPCF) between 2012 and 2006 for NH and nine air traffic regions are based on ECMWF reanalysis data.**

| Region | Δkm | ΔK | 2012/2006 ratio | ECMWF PPCF diff. |
|---|---|---|---|---|
| NH | 0.257 | -1.62 | 1.250 | +3.01 |
| N Atlantic | 0.442 | -2.38 | 1.004 | +10.7 |
| CONUS | 0.298 | -1.94 | 1.053 | +2.03 |
| Europe | 0.181 | -1.00 | 1.181 | -9.60 |
| W Asia | 0.058 | -0.87 | 1.891 | -7.56 |
| E Asia | 0.144 | -1.23 | 1.721 | -8.67 |
| Europe/LA | 0.231 | -1.72 | 0.432 | +18.4 |
| HI/CONUS | 0.459 | -3.35 | 0.975 | -1.18 |
| NE Pacific | 0.625 | -2.82 | 1.084 | +1.08 |
| NW Pacific | 0.794 | -4.89 | 1.002 | -6.26 |



**Table 3: Season-mean retrieved contrail temperature, τ, $D_e$, and average latitude of NH contrail pixels (Mask B) derived from 2006 and 2012 *Terra* and *Aqua* MODIS data.**

*Terra* 2006

| Season | Tcon (K) | | τ | | De (μm) | | Avg Lat (°) |
|--------|-----|-------|-----|-------|-----|-------|------------|
| | Day | Night | Day | Night | Day | Night | |
| DJF | 218.4 | 218.9 | 0.213 | 0.216 | 34.2 | 34.4 | N/A |
| MAM | 219.8 | 220.6 | 0.216 | 0.207 | 34.7 | 35.5 | N/A |
| JJA | 224.4 | 226.0 | 0.238 | 0.220 | 33.3 | 34.4 | N/A |
| SON | 221.1 | 221.7 | 0.217 | 0.223 | 33.3 | 33.4 | N/A |
| Annual | 221.0 | 221.5 | 0.221 | 0.216 | 33.9 | 34.4 | N/A |

*Terra* 2012

| Season | Tcon (K) | | τ | | De (μm) | | Avg Lat (°) |
|--------|-----|-------|-----|-------|-----|-------|------------|
| | Day | Night | Day | Night | Day | Night | |
| DJF | 217.1 | 217.3 | 0.183 | 0.192 | 31.2 | 31.6 | 37.0 |
| MAM | 220.9 | 222.1 | 0.186 | 0.190 | 30.2 | 31.5 | 37.4 |
| JJA | 223.1 | 224.3 | 0.208 | 0.213 | 30.0 | 30.3 | 41.9 |
| SON | 218.1 | 218.3 | 0.187 | 0.195 | 30.8 | 31.2 | 39.9 |
| Annual | 219.9 | 220.5 | 0.190 | 0.196 | 30.3 | 31.1 | 39.0 |

*Aqua* 2006

| Season | Tcon (K) | | τ | | De (μm) | | Avg Lat (°) |
|--------|-----|-------|-----|-------|-----|-------|------------|
| | Day | Night | Day | Night | Day | Night | |
| DJF | 218.7 | 219.2 | 0.214 | 0.218 | 36.1 | 35.9 | 36.5 |
| MAM | 221.5 | 222.2 | 0.215 | 0.204 | 36.6 | 36.6 | 37.4 |
| JJA | 224.3 | 225.1 | 0.232 | 0.219 | 35.4 | 35.4 | 41.2 |
| SON | 219.9 | 219.9 | 0.211 | 0.214 | 35.6 | 35.4 | 41.2 |
| Annual | 221.1 | 221.6 | 0.218 | 0.214 | 35.9 | 35.8 | 39.1 |

*Aqua* 2012

| Season | Tcon (K) | | τ | | De (μm) | | Avg Lat (°) |
|--------|-----|-------|-----|-------|-----|-------|------------|
| | Day | Night | Day | Night | Day | Night | |
| DJF | 217.2 | 217.6 | 0.178 | 0.190 | 33.4 | 33.4 | 36.1 |
| MAM | 220.9 | 221.5 | 0.181 | 0.181 | 33.2 | 33.3 | 37.6 |
| JJA | 223.1 | 223.8 | 0.205 | 0.200 | 31.8 | 32.6 | 41.7 |
| SON | 218.3 | 218.6 | 0.182 | 0.190 | 33.1 | 32.9 | 38.8 |
| Annual | 219.9 | 220.6 | 0.186 | 0.190 | 32.9 | 33.1 | 38.6 |



**Table 4: Relative change in normalized and total contrail radiative forcing between 2006 and 2012 for the Northern Hemisphere.**

| | Normalized CRF (W m$^{-2}$) | | | | | |
|---|---|---|---|---|---|---|
| | 2012 *Terra* | 2006 *Terra* | % diff. | 2012 *Aqua* | 2006 *Aqua* | % diff. |
| SW | -4.7 | -5.6 | -16 | -4.7 | -5.6 | -16 |
| LW | 10.8 | 14.0 | -23 | 11.5 | 14.2 | -19 |
| Net | 6.1 | 8.4 | -27 | 6.8 | 8.5 | -20 |
| | Total CRF (mW m$^{-2}$) | | | | | |
| | 2012 *Terra* | 2006 *Terra* | % diff. | 2012 *Aqua* | 2006 *Aqua* | % diff. |
| SW | -6.3 | -7.2 | -12 | -6.4 | -7.1 | -10 |
| LW | 14.3 | 17.8 | -20 | 15.5 | 17.9 | -13 |
| Net | 8.0 | 10.6 | -24 | 9.03 | 10.7 | -16 |

**Table 5: Relative change in total contrail radiative forcing between 2006 and 2012 for the Northern Hemisphere and nine air traffic regions.**

| Region | *Aqua* SWCRF | *Terra* SWCRF | *Aqua* LWCRF | *Terra* LWCRF | *Aqua* Net CRF | *Terra* Net CRF |
|---|---|---|---|---|---|---|
| NH | -9.8 % | -12.4 % | -13.4 % | -19.9 % | -15.9 % | -24.9 % |
| N Atlantic | -1.4 % | +0.9 % | -3.9 % | -7.0 % | -5.3 % | -14.2 % |
| CONUS | -11.7 % | -13.3 % | -18.6 % | -22.4 % | -23.3 % | -28.0 % |
| Europe | -21.2 % | -19.5 % | -25.5 % | -27.4 % | -28.3 % | -32.2 % |
| W Asia | -3.7 % | -11.9 % | -15.5 % | -20.2 % | -19.8 % | -23.8 % |
| E Asia | -14.3 % | -11.0 % | -8.3 % | -14.7 % | -4.5 % | -16.7 % |
| Europe/LA | -45.8 % | -50.0 % | -48.5 % | -61.3 % | -51.2 % | -66.3 % |
| HI/CONUS | -26.0 % | -18.9 % | -28.7 % | -32.5 % | -31.5 % | -42.7 % |
| NE Pacific | -16.4 % | -22.8 % | -16.2 % | -30.0 % | -16.1 % | -35.4 % |
| NW Pacific | -9.6 % | -19.8 % | -19.3 % | -21.9 % | -23.5 % | -23.2 % |





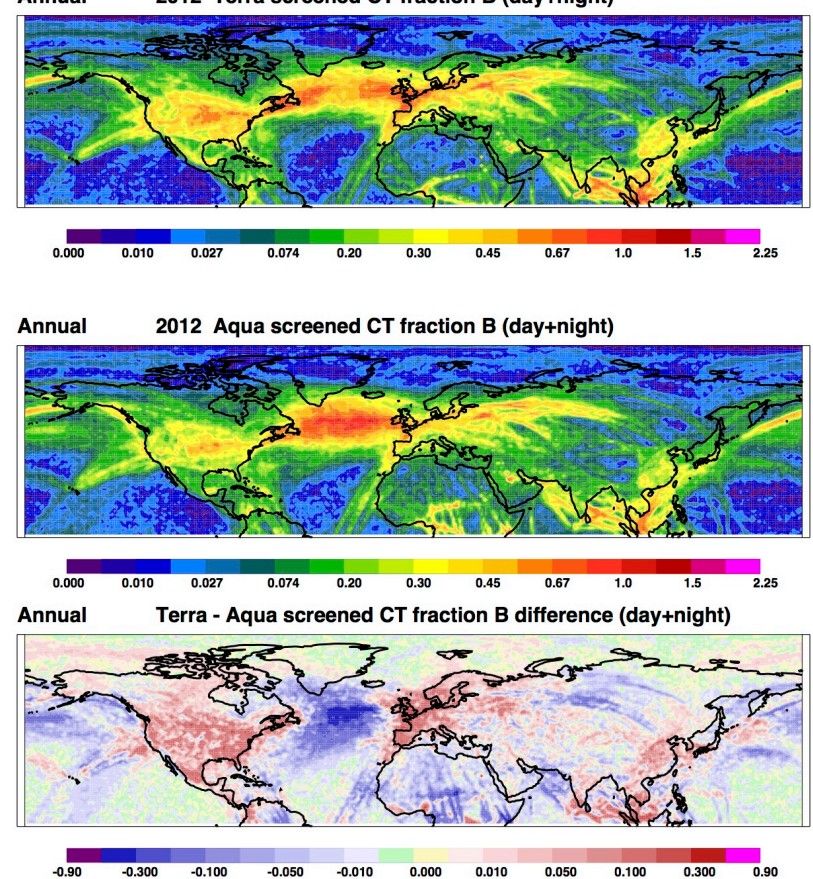

**Figure 1:** **The 2012 annual-mean screened NH CC from *Terra* and *Aqua* MODIS imagery (Mask B), and the**
5 **difference between screened coverage for the two satellites.**



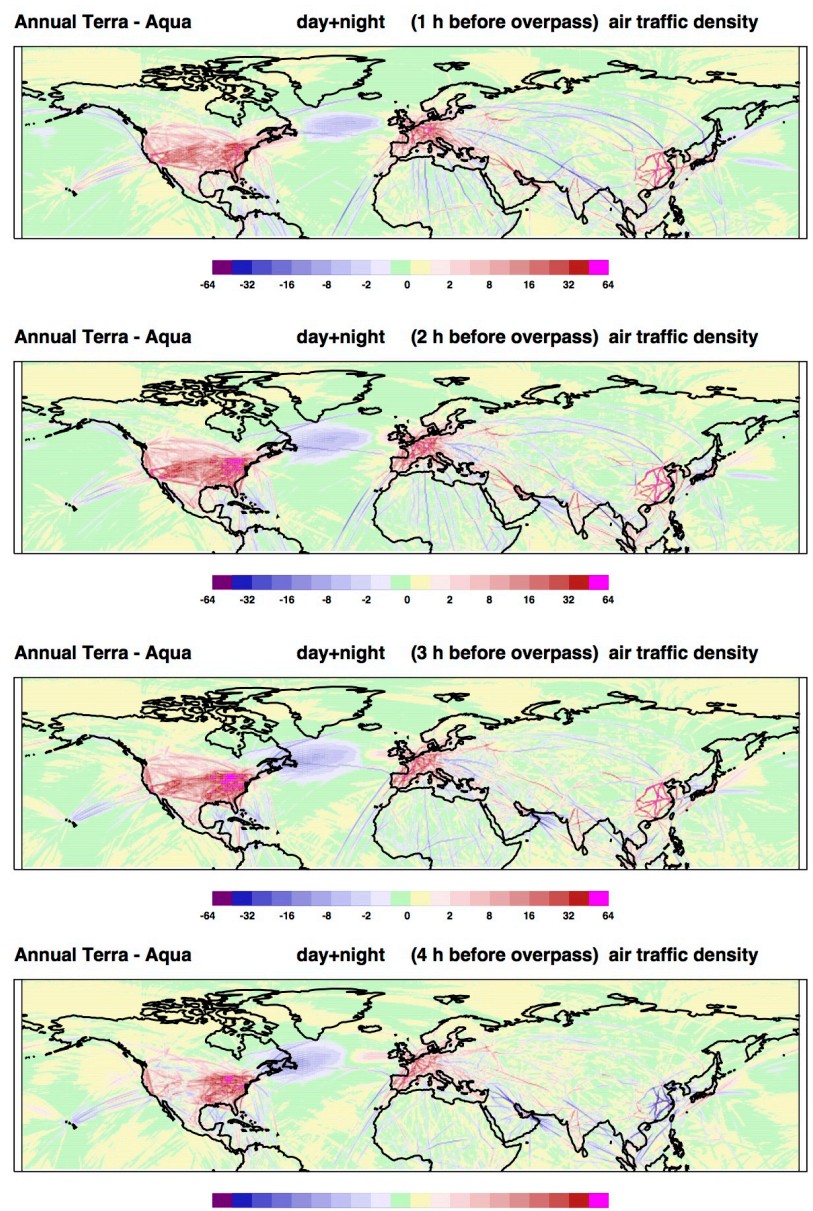

5 **Figure 2: The difference in the 2012 annual-mean air traffic density relative to the *Terra* and *Aqua* MODIS overpass times, for 1-h intervals from 1 to 4 h before overpass.**



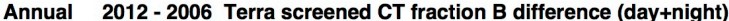

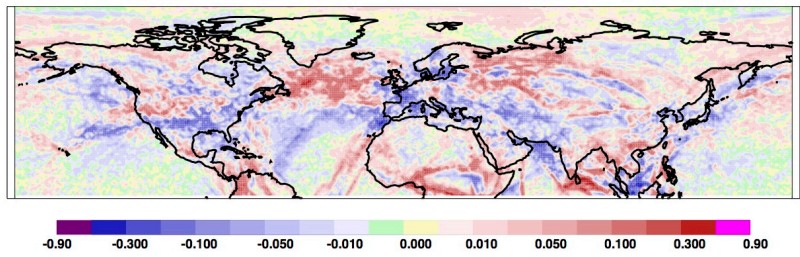

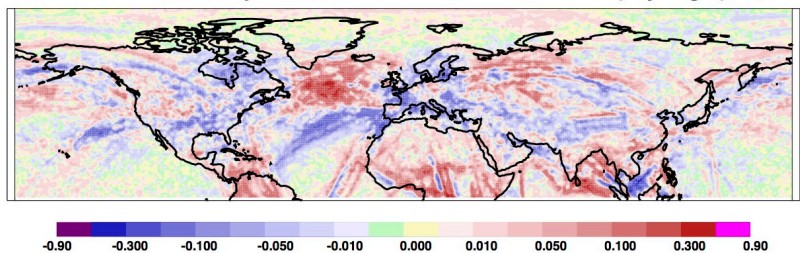

5   **Figure 3: Interannual (2012 minus 2006) difference in the annual-mean screened NH CC from *Terra* and *Aqua* MODIS imagery (Mask B).**





**Annual    2012 - 2006  Terra unscreened CT fraction B difference (day+night)**

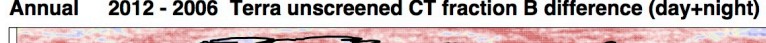

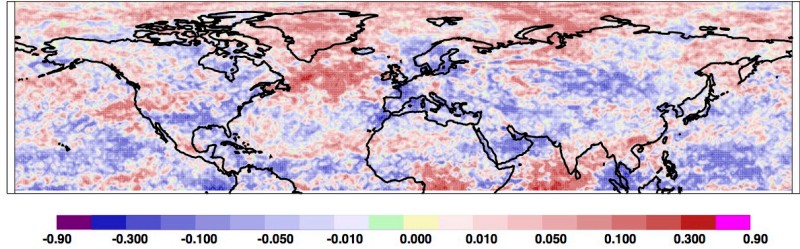

**Annual    2012 - 2006  Aqua unscreened CT fraction B difference (day+night)**

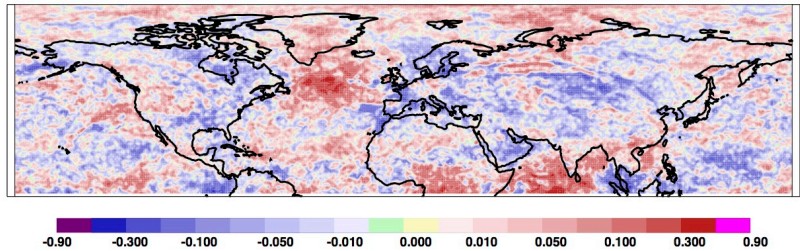

**Figure 4:** **Interannual (2012 minus 2006) difference in the annual-mean unscreened NH CC from *Terra* and *Aqua* MODIS imagery (Mask B).**



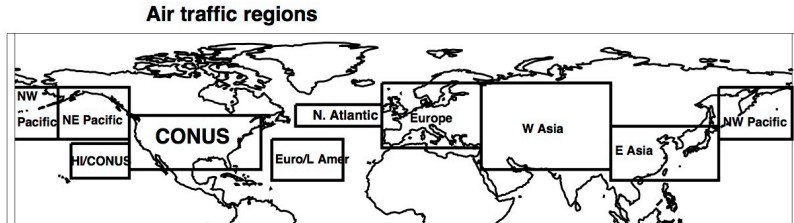

**Figure 5:  Nine air traffic regions selected for study.**



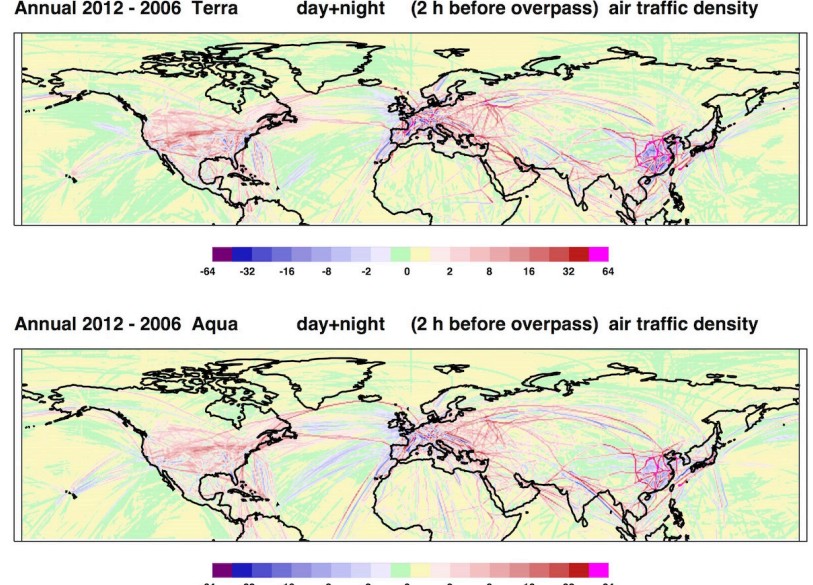

**Figure 6: Interannual (2012 minus 2006) difference in the annual-mean air traffic density relative to 2 h before the**
5 ***Terra* and *Aqua* MODIS overpass times.**





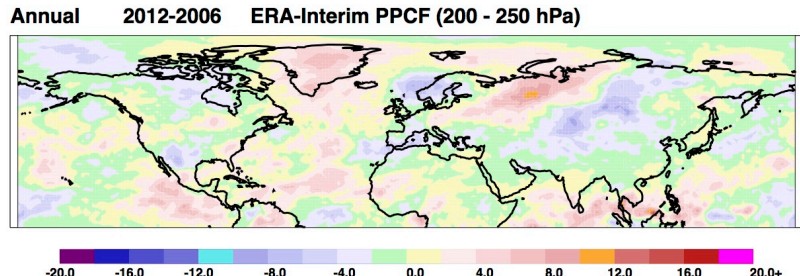

**Figure 7: Interannual (2012 minus 2006) difference in the potential persistent contrail frequency computed between 200 and 250 hPa from ERA-Interim reanalysis data.**





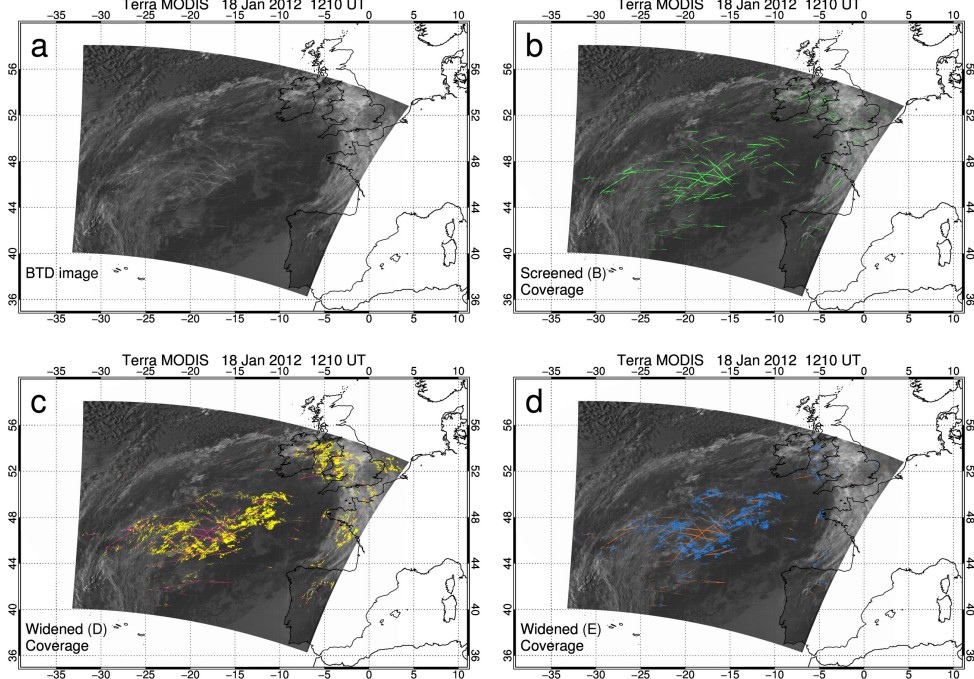

**Figure 8:** **[a] Map of 11 – 12 µm brightness temperature difference (BTD) from 5-minute *Terra* MODIS granule starting at 1210 UT on 18 Jan 2012 over the eastern North Atlantic. [b] Similar to [a], but with Mask B screened coverage in green. [c] Similar to [a], but including Mask D linear contrails in pink and contrail cirrus in yellow. [d] Similar to [c], but containing Mask E linear contrails as orange pixels and contrail cirrus as light blue pixels.**





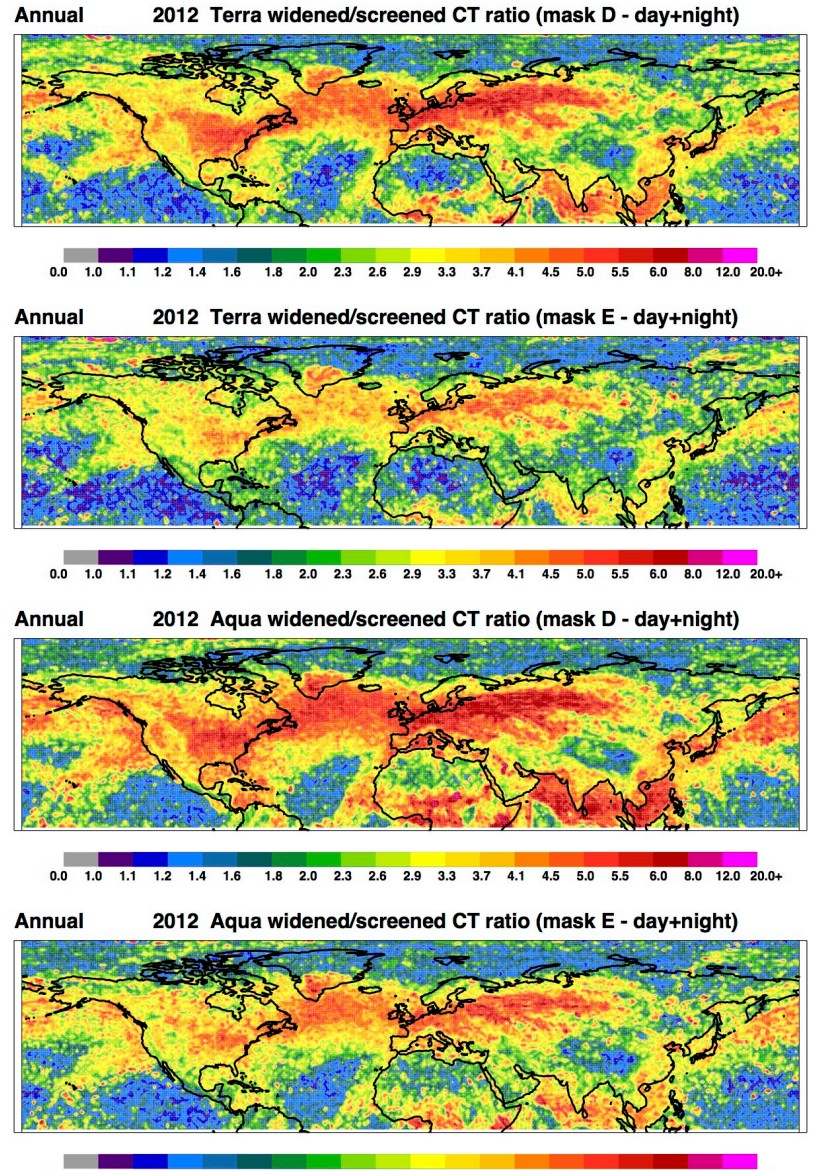

**Figure 9:** The 2012 annual-mean NH widened versus screened CC ratio from *Terra* and *Aqua* MODIS imagery (Masks D and E).





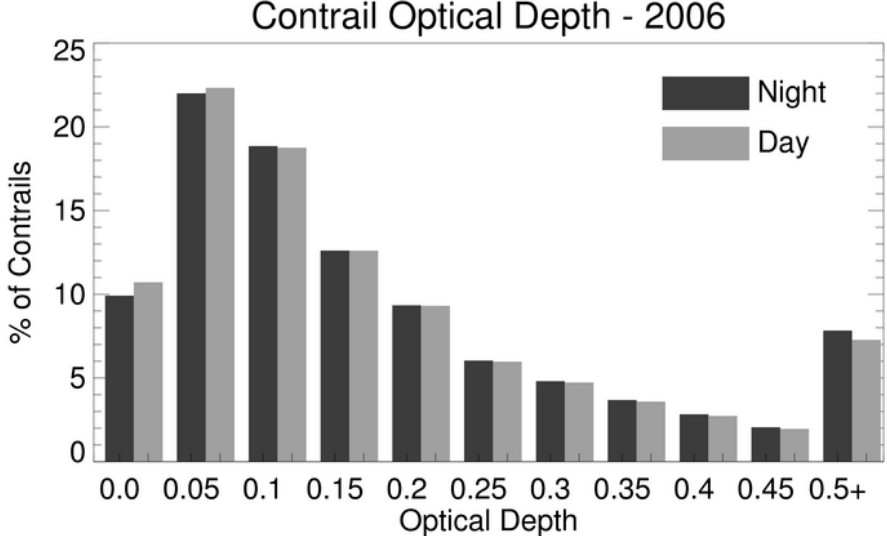

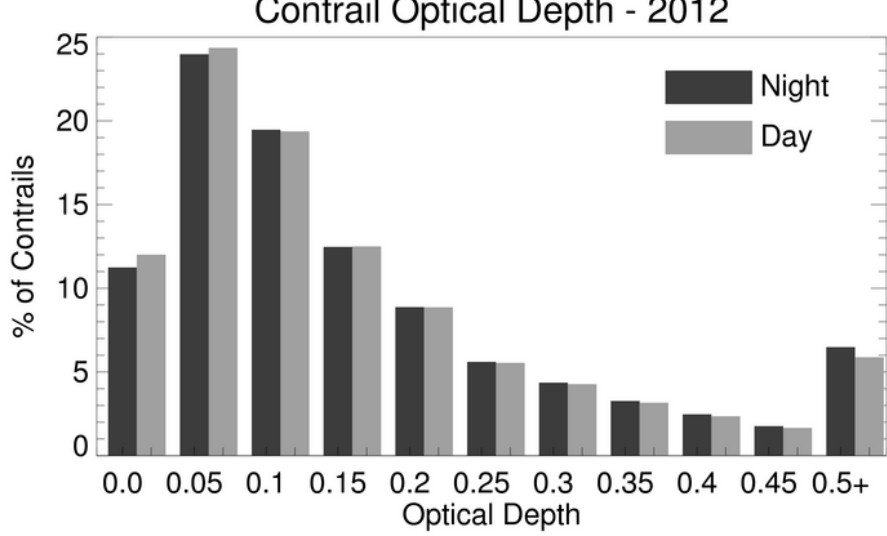

**Figure 10: Histograms of retrieved contrail τ computed from 2006 and 2012 *Terra* MODIS data.**



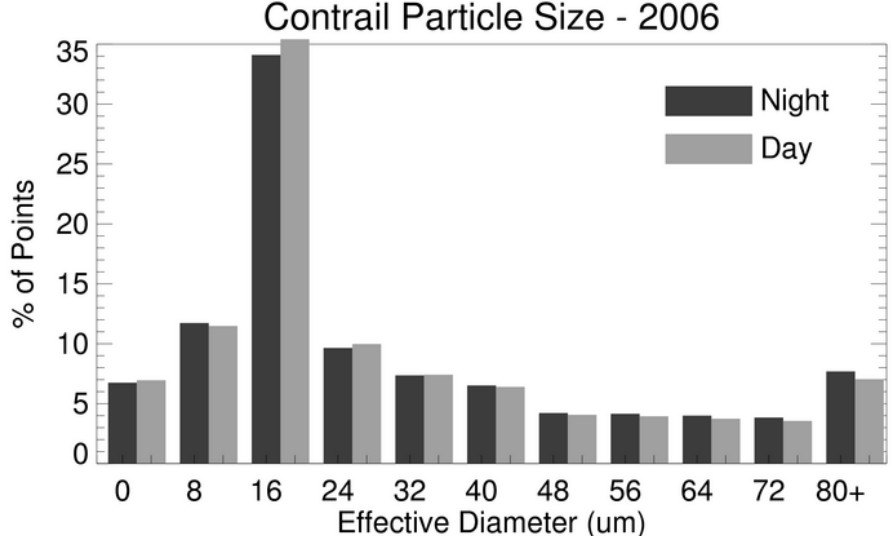

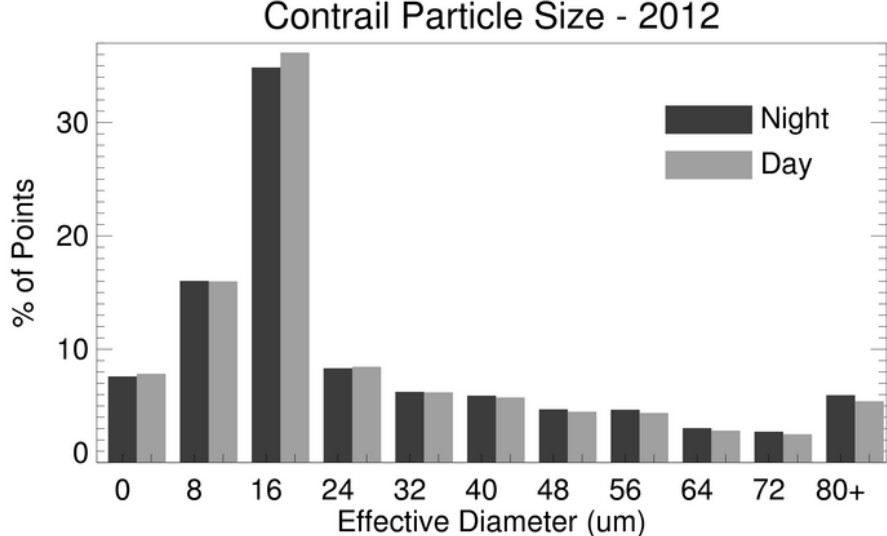

**Figure 11: Histograms of retrieved contrail $D_e$ computed from 2006 and 2012 *Terra* MODIS data.**



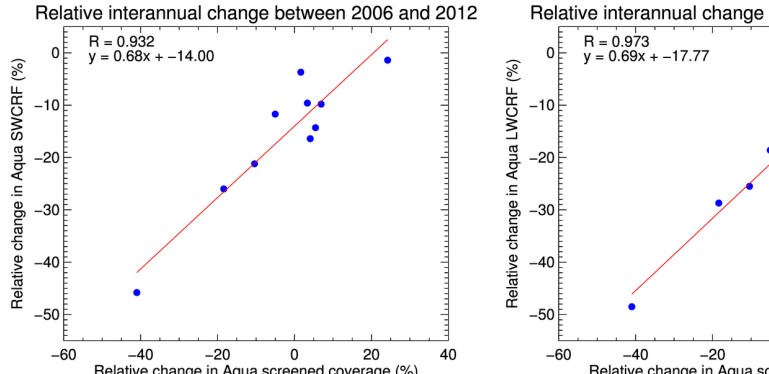

**Figure 12: Relative change ([2012 − 2006]/2006) in the total *Aqua* contrail SWCRF and LWCRF for NH and nine air traffic regions as a function of the relative change in screened CC (day+night, Mask B).**

