# Peer review of "Northern Hemisphere Contrail Properties Derived from Terra and Aqua MODIS Data for 2006 and 2012"

_Atmospheric Chemistry and Physics, 2018_

## Referee Comment (RC1) · Anonymous Referee #1 · 27 Oct 2018

The paper addresses important objectives: Northern Hemisphere (NH) contrail properties, in terms of coverage, optical depth, particles sizes and radiative forcing, and their changes from 2006 to 2012. It uses valuable data at high level of remote sensing expertise: multispectral MODIS data with high spatial and some temporal resolution from two polar orbiting satellites (AQUA and TERRA), from 2 years. It uses an established algorithm which has been shown to be able to detect linear contrails, at least over quasi-homogeneous surfaces (such as the oceans) and for weak traffic where overlap from various contrails and overlap with other clouds is less important.

The method was known to suffer from spatially variable detection efficiencies and from

possibly large false detection rates from misinterpretation of linear structures in natural cirrus. The overpass times of the satellites changed somewhat between the years 2006 and 2012. The changes may have some impact on the results in particular in regions with strong diurnal traffic cycles, such as over the North Atlantic. For correction, meteorological data and traffic data are used, which unfortunately are different in several respect and it is not clear whether the quality of the data over the two observation periods is sufficient to allow for an unbiased comparison of the results from the two one-year periods.

The paper comes to important conclusions: most contrails are 2 h old when detected by the satellites. That conclusion is reasonable and consistent with a few other studies (not only form their own team).

Further the paper suggests that the NH contrail coverage increased from 0.136 % to 0.140 % in coverage or by 3 % in relative terms. Unfortunately, error estimates on these results are missing and difficult. One cannot be sure about the significance of the small changes because that would require an overall accuracy better than 3 %. So, these data should be presented together with error estimates, which may be large. At present the abstract presents the coverage results as if they were accurate to 3 digits. That needs to be changed.

Figure 1 shows the derived annual mean global distribution of detected contrail coverage. The result suggests a strong contrail maximum over the North Atlantic. The result of Figure 1 may be technically correct but the overall result does not look plausible. It contradicts many other studies in terms of the spatial distribution of contrail coverage. See all the global model studies on contrails that have been published so far since 1998 (see reviews in IPCC 1999, 2013, etc.). All of them show contrail maxima over the continents, not over the North Atlantic.

The authors discuss traffic and potential contrail coverage computed for the given traffic using numerical weather prediction data but do not show a NH map of the absolute

values of potential contrail coverage and the product of the potential coverage with traffic for comparisons. Figure 2 only shows differences in these parameters between the two annual periods. I strongly suggest to add a plot of the expected coverage and to point out that Figure 1 suffers from the spatially variable detection efficiency.

I suggest that the paper presents a table for the nine air traffic regions identified in Figure 5, comparing the observed contrail properties (coverage, RF, etc.) with computed or model-estimated contrail properties.

The authors cite Meyer et al. (2002, JGR, doi: 10.1029/2001jd000426) but the list of references misses this paper. Another paper, Meyer et al (2007, Int. J. Rem. Sens., doi: 10.1080/01431160600641707) also discussed contrail coverage, and their Table 1 shows what I was looking for: a comparison between observed and computed contrail coverage over various regions of the world. Of course, nowadays such a comparison can be made far better than >10 years ago, and other model results became available in the literature.

The discussion o altitude changes is not convincing. There is no reasonable and testable argument given for why the mean cruise altitude of air traffic should have been increasing by 0.26 km or 0.79 km over the NH or over the Pacific during the just <6 years since 2006, except that two data sets of different origin indicate this. I suggest skipping this discussion and the related Table 2.

The values given for global radiative forcing do not yet contain error bounds for possibly underestimated contrail coverage over the continents. I suggest that the authors estimate possible underestimates over the continents (e.g., from the mentioned comparison to model data) and use such estimates to derive an upper bound on contrail coverage or RF from their data.

The discussion of "interannual" changes should be reduced. There is no significance in the detected "interannual variability" if only 2 years are considered. The best one could do is to report differences between the two years. So instead of saying the cover

changed from 2006 to 2012, they should say the data from 2012 and 2006 show differences, but should add that the differences can have many reasons, including true contrail changes, humidity changes, traffic changes, changes in the observation method, etc.

I encourage the authors to carefully revise the paper and to publish the facts and the data sets, with proper comparisons to model results, more restrictive conclusions, and self-critical discussion.
* * *

---

## Referee Comment (RC2) · Anonymous Referee #2 · 16 Nov 2018

The study involves a very relevant comparison of satellite contrail retrieval outputs by contrasting annual averages from two years in terms of differences in traffic, coverage, optical depth, and particle size. Nevertheless, this comparison is confounded by differences in altitude, meteorology and background characterisation techniques. I would strongly suggest that all comparisons in the study are performed separately for each variable, while keeping all others constant. I believe that this should be easily done with the data already available in the study, as this would greatly expand the applicability of the results to a wider community.

The title of the article should reflect the fact that this is a comparison of two years of

contrail retrievals with respect to variables not necessarily linked to "interannual variability", as it is the case for traffic and altitude changes between the two years. I would make the following specific suggestions:

a) Provide an estimate of the uncertainties and differences in the calculated potential contrail coverage between the ECMWF and MERRA data. This will allow modellers to inform their choice of data base and help to quantify the uncertainties linked to the calculated contrail coverage. It would be useful to give these differences in PPCF from the ECMWF and MERRA as maps and latitudinal and global averages. Depending on the temporal pattern of the differences, the results might need to be presented as seasonal or monthly averages.

b) It would be useful to complement Table 2 with maps of temperature and PPCF, but in this case contrasting the differences between 2006 and 2012. This will make it easier to understand the latitudinal dependence of PPCF on temperature changes and validate them by screened CC retrievals. The maps, again, should probably correspond to representative seasons or months, depending on their variability between the two years.

c) In order to explain the differences between the two years in terms of the change in altitude, it should be easy with your available data to perform PPCF calculations using the actual altitudes and present them in a map together with traffic differences and their resulting CC. This will provide an observational measure of the relative dependence of CC on altitude. The comparisons will require to first keep traffic volume constant in order to assess the altitude dependence only, and then assess the contribution from traffic volume differences.

I believe that this altitude-dependence assessment will provide extremely useful information to link model outputs and climatological data on how the optical depth and De can be prescribed in terms of ambient temperature, please do not exclude this section from the manuscript.

d) It is not clear to me how contrail radiative forcing was calculated, this should be appropriately described in the manuscript.

e) The suggested analyses should provide a way to discriminate the sources of the differences in retrieved CC between the two years. For these analyses the background characterisation must therefore be somehow be kept constant so it does not affect the conclusions.

I believe that with these additions the paper will make a much more significant contribution to the way in which we understand contrail retrievals from satellites and guide the use of retrieved atmospheric and contrail data in contrail models.

Pg 2 ln 28, delete "and"

---

## Referee Comment (RC3) · Anonymous Referee #3 · 22 Nov 2018

General comments: This paper compare the linear contrail coverage, optical property, and radiative forcing data over the Northern Hemisphere (NH) 2006 and 2012 year of Terra and Aqua MODIS imagery. In the section of Methodology, authors said they employ the optimized CDA algorithm with different contrail masks, while the mid-range Mask B have the best overall balance between falsely detected and missed contrails.

Specific comments: Different with other two Referees, I have such the following specific suggestions: 1. The CDA and modified CDA had made lots of great works, and the detection efficiency raise up all the time. But authors did not release their source code and date sets. It is different to compare their result for other scientists, for example different contrails detection method with the same datasets, or the CDA and modified CDA with other satellite imagery. 2. Two new masks (labeled Mask D and E) were developed to estimate contrail cirrus coverage. Please illustrate the difference among different masks. How the post-processing method detect non-linear contrail cirrus missed by the CDA, wehther could be verified with Geostationary satellite in local region?

3. Author said that the total contrail cirrus coverage visible in the MODIS imagery may be three to four times larger than the linear contrail, how to make sure that.

I suggest the authors could release the source of contrail detection and the data sets, while carefully revise the paper with more comparisons and more restrictive conclusions. With the source code and data sets as supplementary materials , I think more and more scientists will participate the research how the contrails impact radiative forcing, even climate change.

---

## Author Response (AR1)

We thank reviewer #1 again for their review and comments. They have helped to improve our manuscript.

*The paper addresses important objectives: Northern Hemisphere (NH) contrail properties, in terms of coverage, optical depth, particles sizes and radiative forcing, and their changes from 2006 to 2012. It uses valuable data at high level of remote sensing expertise: multispectral MODIS data with high spatial and some temporal resolution from two polar orbiting satellites (AQUA and TERRA), from 2 years. It uses an established algorithm which has been shown to be able to detect linear contrails, at least over quasi-homogeneous surfaces (such as the oceans) and for weak traffic where overlap from various contrails and overlap with other clouds is less important.*

*The method was known to suffer from spatially variable detection efficiencies and from possibly large false detection rates from misinterpretation of linear structures in natural cirrus.*

The claim of "possibly large false detection rates from…natural cirrus" appears to be unsupported speculation by the referee. The manuscript text mentions multiple times that flight tracks are used to screen out false detections. Detection efficiency is discussed later in this reply and dealt with in more detail in a supplement to the paper.

*The overpass times of the satellites changed somewhat between the years 2006 and 2012. The changes may have some impact on the results in particular in regions with strong diurnal traffic cycles, such as over the North Atlantic.*

This statement is not correct. Terra and Aqua were designed to maintain their overpass times and have been kept in their nominal orbits ever since operations began. The reviewer is welcome to check the overpass times using NASA Langley's orbital overpass predictor at

https://cloudsway2.larc.nasa.gov/cgi-bin/predict/predict.cgi

The following table includes the local overpass times at the Equator computed for 2006 and 2012 Terra and Aqua. All of the overpasses are within a few minutes of each other and of the nominal overpass times (1030 LT for Terra, 1330 LT for Aqua).

| Terra | | | | |
|---|---|---|---|---|
| Date | Overpass Latitude | Overpass Longitude | Overpass Time (UT) | Overpass Time (LT) |
| 1 Jan 2006 | 0.0 N | -4.33 W | 10:45:52 | 10:28:34 |
| 1 Apr 2006 | 0.0 N | +11.14 E | 09:44:36 | 10:29:11 |
| 1 Jul 2006 | 0.0 N | -8.98 W | 11:05:21 | 10:29:25 |
| 1 Oct 2006 | 0.0 N | +9.66 E | 09:50:47 | 10:29:25 |
| 31 Dec 2006 | 0.0 N | -10.50 W | 11:11:35 | 10:29:35 |
| 1 Jan 2012 | 0.0 N | +6.46 E | 10:03:43 | 10:29:34 |
| 1 Apr 2012 | 0.0 N | +11.18 E | 09:44:58 | 10:29:41 |
| 1 Jul 2012 | 0.0 N | -8.94 W | 11:05:28 | 10:29:43 |
| 1 Oct 2012 | 0.0 N | +9.63 E | 09:50:58 | 10:29:30 |
| 31 Dec 2012 | 0.0 N | -10.45 W | 11:11:40 | 10:29:52 |
| Aqua | | | | |
| 1 Jan 2006 | 0.0 N | -1.99 W | 13:39:57 | 13:31:58 |
| 1 Apr 2006 | 0.0 N | -11.32 W | 14:17:58 | 13:32:41 |
| 1 Jul 2006 | 0.0 N | -6.65 W | 14:00:33 | 13:33:56 |
| 1 Oct 2006 | 0.0 N | +11.80 E | 12:47:39 | 13:34:50 |
| 31 Dec 2006 | 0.0 N | -8.22 W | 14:07:53 | 13:35:01 |
| 1 Jan 2012 | 0.0 N | +8.80 W | 12:59:46 | 13:34:58 |
| 1 Apr 2012 | 0.0 N | -11.35 W | 14:20:58 | 13:35:34 |
| 1 Jul 2012 | 0.0 N | -6.67 W | 14:02:06 | 13:35:26 |
| 1 Oct 2012 | 0.0 N | +11.93 E | 12:47:26 | 13:35:10 |
| 31 Dec 2012 | 0.0 N | -8.30 W | 14:08:30 | 13:35:18 |

As a test of how consistent the CDA was between the two satellites, we have computed the two-year relative change [(2012 – 2006)/2006×100%] in seasonal [DJF, MAM, JJA, SON] screened and unscreened contrail coverage derived from *Terra* MODIS data versus the corresponding seasonal two-year change in contrail coverage computed from *Aqua* MODIS data for each of the high air traffic regions. The results are plotted below in Figure X, which has been added to the manuscript. Figure X(a) shows a scatter plot of the relative difference in seasonal unscreened contrail coverage between 2012 and 2006 determined from *Terra* MODIS data for each of the high air traffic regions versus the corresponding 2012 minus 2006 relative difference in *Aqua*-derived unscreened coverage. Figure X(b) shows the same scatter plot with the linear regressions for each of the air traffic regions. The unscreened coverages from both satellites are well correlated with each other. The *Terra* and Aqua screened coverages are even better correlated (Figures X(c) and X(d)).

[Figure]

Figure X: Scatter plots of relative difference [(2012 – 2006)/2006×100%] in *Terra* MODIS-derived contrail coverage versus *Aqua* MODIS-derived contrail coverage for each air traffic region.

*For correction, meteorological data and traffic data are used, which unfortunately are different in several respect and it is not clear whether the quality of the data over the two observation periods is sufficient to allow for an unbiased comparison of the results from the two one-year periods.*

The 2006 meteorological data are from GEOS version 4 (which ended in 2007). We expect only minor differences between GEOS-4 and MERRA, the latter being built on GEOS version 5, which was found to have little impact on cloud detection except in polar regions where surface temperatures are different (see Minnis et al. https://ceres.larc.nasa.gov/documents/STM/2007-04/ce0704241020CloudsMinnis.pdf, https://ceres.larc.nasa.gov/documents/STM/2008-05/pdf/3_Minnis.CERES.5.08.pdf). Thus, we do not expect the winds to be significantly different.

*The paper comes to important conclusions: most contrails are 2 h old when detected by the satellites. That conclusion is reasonable and consistent with a few other studies (not only form their own team).*

As indicated in the text, this conclusion is also supported by the results of Vázquez-Navarro et al. (2015).

*Further the paper suggests that the NH contrail coverage increased from 0.136 % to 0.140 % in coverage or by 3 % in relative terms. Unfortunately, error estimates on these results are missing and difficult. One cannot be sure about the significance of the small*

*changes because that would require an overall accuracy better than 3 %. So, these data should be presented together with error estimates, which may be large. At present the abstract presents the coverage results as if they were accurate to 3 digits. That needs to be changed.*

We agree that the uncertainty in the screened contrail coverage estimates are probably large enough that the differences between 2006 and 2012 are not likely to be statistically significant, in large part because we have no way of evaluating the air traffic data that are critical to the screening process. We expect the detectability of contrails from year to year to be the same, but the unscreened data, while consistent with the relative changes in screened data over many regions, indicate no change in coverage. Thus, the small positive increase in screened coverage may not be meaningful.  Please note that the reported coverage changes are small, thus to compute the relative change, we were required to express the coverage with at least three significant digits. We have removed mention of the 3-digit estimates from the abstract, and modified the text to acknowledge the uncertainty in the screened coverage estimates.

*Figure 1 shows the derived annual mean global distribution of detected contrail coverage. The result suggests a strong contrail maximum over the North Atlantic. The result of Figure 1 may be technically correct but the overall result does not look plausible. It contradicts many other studies in terms of the spatial distribution of contrail coverage. See all the global model studies on contrails that have been published so far since 1998 (see reviews in IPCC 1999, 2013, etc.). All of them show contrail maxima over the continents, not over the North Atlantic.*

We remind the reviewer that what we are measuring is not the same as what the models estimate. The satellites detect only some contrails, and at a specific moment in time.  The models simulate all contrails (including contrails in preexisting clouds) and average over time. Also, the North Atlantic is a region favorable for the formation of persistent contrails. We have enclosed our annual-mean estimate of the frequency of persistent contrail formation (in percent) between 250 to 200 hPa (typical aircraft cruise altitudes), based on ERA-Interim reanalysis data. The PPCF is higher over the North Atlantic compared to most of Europe and CONUS.

[Figure]

[Figure]

**Annual   2012   ERA-Interim potential persistent contrail fraction**

| 0.0 | 6.0 | 12.0 | 18.0 | 24.0 | 30.0 | 36.0 | 42.0 | 48.0 | 54.0 | 60.0+ |

*The authors discuss traffic and potential contrail coverage computed for the given traffic using numerical weather prediction data but do not show a NH map of the absolute values of potential contrail coverage and the product of the potential coverage with traffic for comparisons. Figure 2 only shows differences in these parameters between the two annual periods. I strongly suggest to add a plot of the expected coverage and to point out that Figure 1 suffers from the spatially variable detection efficiency.*

Relating the potential and observed contrail coverage would be a good project for another paper, but it would require much additional work to determine where natural cirrus and other high clouds may impact the detection of contrails. We also note that we still do not know exactly how detectable contrail coverage relates to potential coverage and air traffic density.

*I suggest that the paper presents a table for the nine air traffic regions identified in Figure 5, comparing the observed contrail properties (coverage, RF, etc.) with computed or model-estimated contrail properties.*

Such a table would be for another study altogether. To make a **fair** comparison, the model results would have to be screened for natural cirrus and other high ice clouds that would render most contrails invisible to the satellite. In addition, model-based estimates would have to consider the detectability limitations of the satellite imagery and the temporal and spatial sampling of the satellite observations. This would be a far bigger task than due for this paper.

*The authors cite Meyer et al. (2002, JGR, doi: 10.1029/2001jd000426) but the list of references misses this paper. Another paper, Meyer et al (2007, Int. J. Rem. Sens., doi: 10.1080/01431160600641707) also discussed contrail coverage,*

The missing Meyer references have been added to the text.

*and their Table 1 shows what I was looking for: a comparison between observed and computed contrail coverage over various regions of the world. Of course, nowadays such a comparison can be made far better than >10 years ago, and other model results became available in the literature.*

See discussion of model comparisons above.

*The discussion o altitude changes is not convincing. There is no reasonable and testable argument given for why the mean cruise altitude of air traffic should have been increasing by 0.26 km or 0.79 km over the NH or over the Pacific during the just <6 years since 2006, except that two data sets of different origin indicate this. I suggest skipping this discussion and the related Table 2.*

We observe that the technique for reporting heights is the same over CONUS in both datasets, thus the 0.3 km increase in that region is very likely to be real. The differences over other areas may be less certain, but if the CONUS heights are right, the others are probably in the same direction. The height information is used in the retrieved contrail properties (optical depth, effective particle size, and radiative forcing), so we believe it must be included in the paper. To clarify the discussion in the text, we have added that the change in mean cruise altitude was a reported change.

*The values given for global radiative forcing do not yet contain error bounds for possibly underestimated contrail coverage over the continents. I suggest that the authors estimate possible underestimates over the continents (e.g., from the mentioned comparison to model data) and use such estimates to derive an upper bound on contrail coverage or RF from their data.*

This study focuses on our satellite-based estimates. An estimate of RF over the continents has already been included in the paper based on the contrail cirrus estimate. We also have unpublished work that can provide an annual-mean corrected (for background inhomogeneity) contrail coverage estimate for the 2006 data. This estimate is based on a visual analysis that we performed on the 2006 data, including an inhomogeneity correction based on Meyer et al. (2002). A brief summary of that work has been included as a supplement. The correction increases the Terra NH-mean coverage by 25 percent (from 0.136 to 0.170%), which would thus increase the overall linear contrail RF by an equal proportion. (The corrected 2006 Aqua coverage is 0.169%.) The corrected Terra coverage shows that the maxima in CC is now over CONUS and Europe, although the corrected Aqua coverage still has a maximum over the North Atlantic due to the decrease in detectability of linear contrails over the continents during the afternoon. We are not able to re-do the visual analysis for the 2012 coverage due to the considerable labor requirements. Assuming the same correction as for 2006, the corrected 2012 Terra (Aqua) coverage increases to 0.178% (0.185%).

[Figure]

Annual 2006 Terra — Corrected CT fraction (three step method)

Annual 2006 Aqua — Corrected CT fraction (three step method)

*The discussion of "interannual" changes should be reduced. There is no significance in the detected "interannual variability" if only 2 years are considered. The best one could do is to report differences between the two years. So instead of saying the cover changed from 2006 to 2012, they should say the data from 2012 and 2006 show differences, but should add that the differences can have many reasons, including true contrail changes, humidity changes, traffic changes, changes in the observation method, etc.*

As the first sentence of the abstract states, this study compares contrail properties derived from satellite data measured during 2006 and 2012. We believe it is clear to the reader that we are discussing changes in observed contrail properties between the two years, and that several factors may be causing the differences. The use of language such as "contrail coverage changed from 2006 to 2012" is simply a report of a difference between results from two years, not a declaration of absolute accuracy in measurement. As for the use of "interannual", we have removed some unnecessary instances of the word (mostly in the figure captions), and used other phrases when possible to describe "interannual". We note that the phrase "interannual variability" only occurs once in the manuscript, where the future possibility of adding an additional year of contrail properties is discussed.

*I encourage the authors to carefully revise the paper and to publish the facts and the data sets, with proper comparisons to model results, more restrictive conclusions, and self-critical discussion.*

The referee's comments here are not clear to us. What is meant by "publish the facts and the data sets"? What facts and data sets, the entire two-year, two-satellite set of MODIS imagery? A release of the source code and data sets is not

reasonable. It is not feasible to upload the hundreds of gigabytes of satellite data processed in this study. The source code is experimental and not easily implemented by someone unfamiliar with the programs. In addition to contrail detection, we also retrieve contrail optical properties and radiative forcing with additional code and processing systems.
We thank reviewer #2 for their review and comments. They have helped to improve our manuscript.

*The study involves a very relevant comparison of satellite contrail retrieval outputs by contrasting annual averages from two years in terms of differences in traffic, coverage, optical depth, and particle size. Nevertheless, this comparison is confounded by differences in altitude, meteorology and background characterisation techniques. I would strongly suggest that all comparisons in the study are performed separately for each variable, while keeping all others constant. I believe that this should be easily done with the data already available in the study, as this would greatly expand the applicability of the results to a wider community.*

*The title of the article should reflect the fact that this is a comparison of two years of contrail retrievals with respect to variables not necessarily linked to "interannual variability", as it is the case for traffic and altitude changes between the two years. I would make the following specific suggestions:*

Following the suggestion of anonymous referee #1, we have already revised the title of the paper to "Northern Hemisphere Contrail Properties Derived from Terra and Aqua MODIS Data for 2006 and 2012".

*a) Provide an estimate of the uncertainties and differences in the calculated potential contrail coverage between the ECMWF and MERRA data. This will allow modellers to inform their choice of data base and help to quantify the uncertainties linked to the calculated contrail coverage. It would be useful to give these differences in PPCF from the ECMWF and MERRA as maps and latitudinal and global averages. Depending on the temporal pattern of the differences, the results might need to be presented as seasonal or monthly averages.*

Currently, any estimate of linear contrail coverage using the PPCF from the two meteorological re-analyses and air traffic would be very uncertain. Although the annually-averaged spatial patterns of PPCF calculated from ERA-Interim and MERRA are generally similar, the absolute values differ by nearly a factor of two. (Notice the difference in the scale in the following annual means.)

[Figure]

**Annual  2006  ERA-Interim potential persistent contrail fraction**

| 0.0 | 6.0 | 12.0 | 18.0 | 24.0 | 30.0 | 36.0 | 42.0 | 48.0 | 54.0 | 60.0+ |

**Annual  2006  MERRA PPCF (200 - 250 hPa)**

| 0.0 | 3.0 | 6.0 | 9.0 | 12.0 | 15.0 | 18.0 | 21.0 | 24.0 | 27.0 | 30.0+ |

It would complicate an already long manuscript to present maps of the monthly or seasonally varying PPCF in the paper. However, we agree that a comparison of the year-to-year variation of seasonal PPCF from both re-analyses with the corresponding changes in contrail coverage would be valuable. To simplify the analysis, we computed the two-year relative change [(2012 – 2006)/2006×100%] in seasonal [DJF, MAM, JJA, SON] screened and unscreened contrail coverage versus the corresponding seasonal two-year absolute (2012 - 2006) change in PPCF computed from both MERRA and ERA-Interim data.  The year-to-year changes in coverage were calculated for each season in each of the nine high air traffic regions plus the NH and plotted versus the corresponding changes in PPCF.

The following figures (Y and Z) have been added to the manuscript. Figure Y(a) shows a scatter plot of the relative difference in seasonal unscreened contrail coverage between 2012 and 2006 determined from *Terra* MODIS data for each of the high air traffic regions versus the corresponding 2012 minus 2006 absolute difference in PPCF computed for each season and each air traffic region from the MERRA re-analysis data. Figure Y(b) shows the same scatter plot with the linear regressions for each of the air traffic regions. Note two outlier plots: the red crosses represent the North Atlantic region while the brown triangle regression with the anti-correlation between coverage and PPCF represents the NE Pacific region. In Figures Y(c) and Y(d), the screened coverage and MERRA PPCF are essentially uncorrelated due to the additional outlier relationships between screened coverage and PPCF (red triangles represent Europe/Latin America corridor; green triangles represent HI/CONUS corridor). Figure Z shows similar relationships between the seasonal *Terra* MODIS-derived contrail coverage and

[Figure]

Figure Y: Scatter plots of *Terra* MODIS-derived contrail coverage versus PPCF computed from MERRA re-analyses of the upper troposphere (150 – 400 hPa).

[Figure]

Figure Z: Scatter plots of *Terra* MODIS-derived contrail coverage versus PPCF computed from ECMWF re-analyses of the upper troposphere (150 – 400 hPa).

the ERA-Interim-based PPCF, although the correlations are stronger than for the MERRA data. Overall, the correlations are better for the PPCFs computed from the ERA-Interim re-analyses, and for the unscreened coverage. The differences between the unscreened coverage and the screened coverage scatter plots suggest that the nature of the air traffic data between 2006 and 2012 may have changed for the Europe/Latin America and the HI/CONUS air routes. The North Atlantic air route appears to be an outlier from the other air traffic regions in both the screened and unscreened contrail coverage scatter plots. Because very few unscreened contrails in the North Atlantic region are screened out by the flight track screening, the similarity between the unscreened and screened results would be expected. The shift of the North Atlantic regression to the right of the other regions suggests that contrails might have been more easily detected in the North Atlantic during 2012. The standard deviation of the background 12-μm brightness temperature, which is known to affect the detectability of linear contrails by the CDA, decreased by about 10% in 2012 compared to 2006 in the North Atlantic region, which may account for some of the discrepancy. (In contrast, however, the HI/CONUS region had a decrease in the 12-μm BT variability of 11 – 15% between 2006 and 2012, but the unscreened coverage changes are in more agreement with the other air traffic regions. The other air traffic regions generally had background heterogeneity changes of less than 5% between the two years.) In addition, the magnitude of the discrepancy between the North Atlantic and the other air traffic regions is noticeably larger in the MERRA-based plots. It appears that there is greater uncertainty between the MERRA- and ECMWF- derived PPCF in this region between 2006 and 2012 than in other regions.

*b) It would be useful to complement Table 2 with maps of temperature and PPCF, but in this case contrasting the differences between 2006 and 2012. This will make it easier to understand the latitudinal dependence of PPCF on temperature changes and validate them by screened CC retrievals. The maps, again, should probably correspond to representative seasons or months, depending on their variability between the two years.*

Please note that the contrail temperatures used in the contrail property retrievals are based on annual means that relate the average contrail altitude/pressure height with temperature. Thus, the inclusion of monthly or seasonal maps of temperature changes is detail beyond what we intended for this study.

*c) In order to explain the differences between the two years in terms of the change in altitude, it should be easy with your available data to perform PPCF calculations using the actual altitudes and present them in a map together with traffic differences and their resulting CC. This will provide an observational measure of the relative dependence of CC on altitude. The comparisons will require to first keep traffic volume constant in order to assess the altitude dependence only, and then assess the contribution from traffic volume differences.*

*I believe that this altitude-dependence assessment will provide extremely useful information to link model outputs and climatological data on how the optical depth and De can be prescribed in terms of ambient temperature, please do not exclude this section from the manuscript.*

The determination of how CC relates to altitude/pressure is not clear from the data. When we plotted the two-year differences in contrail coverage (both screened and unscreened) for each season and each air traffic region with the corresponding two-year change in PPCF *at each pressure level*, none of the plots showed a strong correlation between CC and (one pressure level) PPCF. A stronger correlation was only evident when we used the two-year change in PPCF computed throughout the upper troposphere (150 – 400 hPa). This result at least shows that the relation between satellite-observed CC and re-analysis-derived PPCF with altitude is complicated and the topic of another study.

*d) It is not clear to me how contrail radiative forcing was calculated, this should be appropriately described in the manuscript.*

Text has been added to section 2.3 to describe the calculation of the contrail radiative forcing.

*e) The suggested analyses should provide a way to discriminate the sources of the differences in retrieved CC between the two years. For these analyses the background characterisation must therefore be somehow be kept constant so it does not affect the conclusions.*

*I believe that with these additions the paper will make a much more significant contribution to the way in which we understand contrail retrievals from satellites and guide the use of retrieved atmospheric and contrail data in contrail models.*

It is not clear what the reviewer is requesting here in terms of "background characterization". We agree that the reviewer's suggestions are helpful in minimizing the unavoidable effects that result from having to use some different data sets in the two years of analysis.

*Pg 2 ln 28, delete "and"*

The extra "and" has been deleted.
We thank reviewer #3 for their review and comments. They have helped to improve our manuscript.

*General comments: This paper compare the linear contrail coverage, optical property, and radiative forcing data over the Northern Hemisphere (NH) 2006 and 2012 year of Terra and Aqua MODIS imagery. In the section of Methodology, authors said they employ the optimized CDA algorithm with different contrail masks, while the mid-range Mask B have the best overall balance between falsely detected and missed contrails.*

*Specific comments: Different with other two Referees, I have such the following specific suggestions: 1. The CDA and modified CDA had made lots of great works, and the detection efficiency raise up all the time. But authors did not release their source code and date sets. It is different to compare their result for other scientists, for example different contrails detection method with the same datasets, or the CDA and modified CDA with other satellite imagery.*

A release of the source code and data sets is not reasonable. Relatively few contrail detection papers have been published due to the difficulty in processing such large satellite datasets. It is not feasible to upload the hundreds of gigabytes of satellite data processed in this study. The source code is experimental and not easily implemented by someone unfamiliar with the programs. In addition to contrail detection, we also retrieve contrail optical properties and radiative forcing with additional code and processing systems.

*2. Two new masks (labeled Mask D and E) were developed to estimate contrail cirrus coverage. Please illustrate the difference among different masks. How the post-processing method detect non-linear contrail cirrus missed by the CDA, wehther could be verified with Geostationary satellite in local region?*

An example of Mask D and E is presented in Figure 11. A description of the post-processing method and the reasoning used to estimate contrail cirrus coverage appears in Section 2.1. As described in the text, visual analysis by a human observer of several MODIS granules was used to verify and to optimize the post-processing method. The visual analysis was limited due to the labor-intensive nature of the assessment, which required several rounds of analysis while the post-processing method was developed. We expect that loops of geostationary satellite data would be helpful in future development of the contrail cirrus mask, but this

would require another study altogether.

*3. Author said that the total contrail cirrus coverage visible in the MODIS imagery may be three to four times larger than the linear contrail, how to make sure that.*

The total contrail cirrus coverage estimate is based on the results of Masks D and E. The assessment of contrail cirrus remains an open problem and requires additional study. We have already included text in the manuscript explaining that the estimates are preliminary and require additional refinement. In the final section of the paper, we have proposed how the contrail cirrus estimates may be improved by using loops of geostationary data to define contrail cirrus coverage better.

*I suggest the authors could release the source of contrail detection and the data sets, while carefully revise the paper with more comparisons and more restrictive conclusions. With the source code and data sets as supplementary materials , I think more and more scientists will participate the research how the contrails impact radiative forcing, even climate change.*

Please see the comment above regarding the release of the satellite data sets.

**Northern Hemisphere Contrail Properties Derived from Terra and Aqua MODIS Data for 2006 and 2012**

David P. Duda[1], Sarah T. Bedka[1], Patrick Minnis[1], Douglas Spangenberg[1], Konstantin Khlopenkov[1], Thad Chee[1], and William L. Smith, Jr.[2]

[1]Science Systems and Applications, Inc., Hampton, VA 23666-5986, USA
[2]NASA Langley Research Center, Hampton, VA 23681-2199, USA

*Correspondence to*: David P. Duda (david.p.duda@nasa.gov)

**Abstract.** Linear contrail coverage, optical property, and radiative forcing data over the Northern Hemisphere (NH) are derived from a year (2012) of *Terra* and *Aqua* Moderate-resolution Imaging Spectroradiometer (MODIS) imagery, and compared with previously published 2006 results (Duda et al., 2013; Bedka et al., 2013; Spangenberg et al., 2013) using a consistent retrieval methodology. Differences in the observed *Terra*-minus-*Aqua* screened contrail coverage and patterns in the 2012 annual-mean air traffic estimated with respect to satellite overpass time suggest that most contrails detected by the contrail detection algorithm (CDA) form approximately 2 h before overpass time.  The 2012 screened NH contrail coverage (Mask B) shows a relative 3% increase compared to 2006 data for *Terra* and increases by almost 7% for *Aqua*, although the differences are not expected to be statistically significant. 
[revised manuscript text omitted]

$$CRF = F_{conf} - F_{con}, \tag{1}$$

where $F_{conf}$ and $F_{con}$ are the upward top-of-atmosphere shortwave or longwave fluxes for contrail-free and contrail-covered conditions, respectively. The fluxes are computed using the Fu-Liou RTM (Fu and Liou, 1993; Fu et al., 1998). $F_{con}$ is derived assuming an atmosphere with contrail-covered conditions where a contrail layer and background cloud, if applicable, are inserted at the relevant altitudes.

$F_{conf}$ is computed for the contrail-free situation for the same background conditions but without the contrail layer. Both the solar and longwave CRF are computed using equation (1), as well as the net CRF, which is the sum of solar and longwave CRF. The contrail properties, $\tau$, $D_e$, and contrail temperature, define the contrail layer for each pixel calculation. Further details of the CRF calculation are discussed in 
[revised manuscript text omitted]

For more insight into the relationship between the detected CC and PPCF, both the CC and PPCF data were sorted by air traffic region and season [DJF, MAM, JJA, SON] for both 2006 and 2012 and compared. As a test of how consistent the detection of CC by the CDA was between both satellites for both years, we first computed the two-year relative change ([2012 – 2006]/2006×100%) in seasonal screened and unscreened contrail coverage derived from *Terra* MODIS data versus the corresponding change in contrail coverage computed from *Aqua* MODIS data for each of the high air traffic regions. Figure 8a shows a scatter plot of the relative difference in seasonal unscreened *Terra* contrail coverage between 2012 and 2006 data for each region versus the corresponding two-year difference in *Aqua*-derived unscreened coverage. Figure 8b shows the same scatter plot with the individual linear regressions for each of the air traffic regions. The unscreened coverage changes from both satellites are well correlated with each other. The year-to-year changes in *Terra* and Aqua screened coverage are even better correlated (Figures 8c and 8d).

Similar scatter plots of the two-year relative change in unscreened and screened CC from *Terra* compared to the corresponding seasonal two-year absolute (2012 - 2006) change in PPCF computed from both MERRA and ERA-Interim data are shown in Figures 9 and 10, respectively. Figure 9a shows a scatter plot of the year-to-year change in *Terra* unscreened CC versus the MERRA-derived PPCF for

each season and each air traffic region.  Figure 9b shows the same scatter plot with the individual linear regressions for each of the air traffic regions. Note two outlier regressions in the plot: the red crosses represent the North Atlantic region while the brown triangle regression with the anti-correlation between coverage and PPCF represents the NE Pacific region. In Figures 9c and 9d, the screened coverage and MERRA PPCF are essentially uncorrelated due to the additional outlier relationships between screened coverage and PPCF (red triangles represent Europe/Latin America corridor; green triangles represent HI/CONUS corridor). Figure 10 shows similar relationships between the seasonal *Terra* MODIS-derived contrail coverage and the ERA-Interim-based PPCF, although the correlations are stronger than for the MERRA data. Overall, the correlations are higher for the PPCFs computed from the ERA-Interim reanalyses, and for the unscreened coverage. The differences between the unscreened coverage and the screened coverage scatter plots in Figures 9 and 10 suggest that the nature of the air traffic data between 2006 and 2012 may have changed for the Europe/Latin America and the HI/CONUS air routes. The North Atlantic air route appears to be an outlier from the other air traffic regions in both the screened and unscreened contrail coverage scatter plots. Because very few unscreened contrails in the North Atlantic region are screened out by the flight track screening, the similarity between the unscreened and screened results is expected. The shift of the North Atlantic regression to the right of the other regions suggests that contrails might have been more easily detected in the North Atlantic during 2012 compared to 2006. The standard deviation of the background 12-μm brightness temperature, which is known to affect the detectability of linear contrails by the CDA (Mannstein et al., 1999), decreased by about 10% in 2012 compared to 2006 in the North Atlantic region, which may account for some of the discrepancy. In addition, the magnitude of the discrepancy between the North Atlantic and the other air traffic regions is noticeably larger in the MERRA-based plots (Figures 9b and 9d) than in the ECMWF plots (Figures 10b and 10d). It appears that there is greater uncertainty between the MERRA- and ECMWF- derived PPCF in this region between 2006 and 2012 than in other regions.

The results presented here demonstrate that interannual changes in air traffic density and PPCF appear to have some influence on the change in screened satellite-detected CC between 2006 and 2012, although some of the changes between the two years are more difficult to explain, especially the large increase in screened and unscreened coverage over the North Atlantic air corridor. The increase in CC between 2006 and 2012 over the North Atlantic may be due to changes in flight altitudes in 2012 that shifted more flights into levels of the atmosphere where ambient conditions are more likely for persistent contrail formation than in 2006. Overall, the uncertainty in the screened contrail coverage estimates are probably large enough that the differences between 2006 and 2012 are not likely to be statistically significant, in large part because of the difficulty in evaluating the air traffic data that are critical to the screening process.

[revised manuscript text omitted]
. Because this report was an analysis and description of our satellite-based contrail observation system, no comparison with model estimates of contrail coverage was included. More research is needed to compare satellite observations of contrails with such modelling estimates, but such a study would require much additional work to determine how satellite-detectable contrail coverage relates to estimates of potential coverage from meteorological conditions and air traffic density. To make a fair comparison, the model results would have to be screened for natural cirrus and other high ice clouds that would render most contrails invisible in the satellite imagery. In addition, model-based estimates would have to consider the overall detectability limitations of the imagery and the temporal and spatial sampling of the satellite observations.

[revised manuscript text omitted]

Lee, D, S., Pitari, G., Grewe, V., Gierens, K., Penner, J. E., Petzold, A., Prather, M. J., Schumann, U., Bais, A., Berntsen, T., Iachetti, D., Lim, L. L., and Sausen, R., Transport impacts on atmosphere and climate: aviation, Atmos. Environ., 44, 4678–4734, 2010.

Mannstein, H., Meyer, R., and Wendling, P., Operational detection of contrails from NOAA-AVHRR data, Int. J. Remote Sensing, 20, 1641–1660, 1999.

Mannstein, H., and Schumann, U., Aircraft induced contrail cirrus over Europe, Meteorologische Zeitschrift, 14(4), 549-554, 2005.

Meerkötter, R., Schumann, U., Doelling, D. R., Minnis, P., Nakajima, T., and Tsushima, Y., Radiative forcing by contrails, Ann. Geophys., 17, 1080-1094, doi:10.1007/s00585-999-1080-7, 1999.

Meyer, R., Mannstein, H., Meerkötter, R., Schumann, U., and Wendling, P., Regional radiative forcing by line-shaped contrails derived from satellite data, J. Geophys.

Res. 107(D10), 4104, doi:10.1029/2001JD000426, 2002.

Meyer, R., Buell, R., Leiter, C., Mannstein, H., Pechtl, S., Oki, T., and Wendling, P., Contrail observations over Southern and Eastern Asia in NOAA/AVHRR data and comparisons to contrail simulations in a GCM, Int. J. Remote Sens., 28(9), 2049–2069, doi:10.1080/01431160600641707, 2007.

[revised manuscript text omitted]

*Terra* **MODIS data.**

[Figure]

**Figure 15: Relative change ([2012 − 2006]/2006 × 100%) in the total *Aqua* contrail SWCRF and LWCRF for NH and nine air traffic regions as a function of the relative change in screened CC (day+night, Mask B).**

---

## Author Response (AR2)

**Co-Editor Decision: Publish subject to minor revisions (review by editor)** (13 Feb 2019) by Matthias Tesche

*Comments to the Author: Dear David,*

**thank you for providing your replies to the reviewers' comments. Your replies have been adequate and addressed most of the issues raised by the reviewers. I agree that it is unnecessary for you to provide the MODIS data used in this study as they are already available elsewhere. I can also understand your reservations regarding the publication of your retrieval code - particularly as it is still experimental and likely not documented to a degree that would enable its application by untrained users. Instead, you are now providing a description of the retrieval and analysis procedure as supplementary material. This is certainly a good addition and will allow the reader to better follow your work.**

I have read your revised manuscript. Your have mentioned yourselves that this is already a long manuscript. I therefore have some comments that I would like you to consider before uploading a final version of your work:

- It is somewhat confusing for the reader to follow when you are shifting focus between different products, i.e. screened and unscreened contrails. For the sake of clarity, I would propose to clearly state in the beginning that your are using Mask B from your earlier work and that contrail detection has been confirmed with the help of aircraft waypoint data. This product provides the highest quality of contrail detection (also in light of the supplementary material) and whatever comes out of it is what is considered a contrail in this work. I would then stick with this product as the focus of your work (there would be no more need to call it screened contrails henceforth) and omit everything that is related to other products, i.e. unscreened contrails. I might have missed something but it doesn't make sense to me to discuss a product that you know to be of inferior quality. I think that the change in air traffic is properly represented in your analysis of the waypoint data in Figs. 2 and 6.

We agree that much of the discussion about unscreened contrail coverage might be confusing and have removed most references to unscreened coverage from section 3.1. However, it is valuable to compare the screened and unscreened coverage as it gives us some insight into how the reported air traffic affects our CC estimates. Thus, we have kept the references to unscreened coverage in Table 1.

**- Please provide a reference to the supplement in your methodology section**

A reference to the supplement has been added to the methodology section, as well as a statement (placed immediately before the "Author contributions" section) indicating that the supplement is available online at the specified web address (the current address is simply a place holder, please amend address as necessary).

- Could you add a short description of the aircraft waypoint data? It is simply a data base of lat, lon, height, speed, etc. for individual aircraft?

The waypoint data are an inventory of the waypoints (latitude, longitude, altitude) of individual reported commercial aircraft flights for 2012. The data have been subjected to a quality control process to remove duplicate data and other data quality problems. For this study, only waypoints at typical contrail formation altitudes ( $\geq$  7.62 km [25 kft]) were included. This description has been added to the manuscript in the methodology section.

- Please provide units for all your plots

Units have been added to all figures where appropriate.

- It is not easy to understand what is shown in Figure 2. I understand that it shows the flight routes that were used at several time intervals before the MODIS observations. But what is meant with annual difference? Is Figure 2b simply the difference between the plots in Figure 6?

Figure 2 shows the air traffic differences relative to each satellite's overpass times (Terra and Aqua) during 2012 (inter-satellite differences in a single year), while Figure 6 shows the air traffic differences between 2006 and 2012 for each satellite (inter-annual differences in a single satellite). We have added some text in Section 3.1 to clarify what is shown in both figures.

- As explained in my first comment, I suggest to omit Figure 4 and any text related to unscreened contrails. Unscreened basically means unverified contrails if I am not mistaken.

We agree with the editor's suggestion and have eliminated Figure 4, in light of the removal of much of the unscreened contrail coverage discussion in the results section.

- Please add a description of the symbols to Figures 8 and 10. Following my earlier comments, data for unscreened contrails should not be show in these figures.

The plots with unscreened contrail data have been removed from both figures. We have added a legend to both figure captions that describes the symbols used in the figures: NH – blue crosses, N Atl – red crosses, CONUS – green crosses, Euro – dark yellow crosses, W Asia – purple crosses, E Asia – blue triangles, Euro/LA – red triangles, HI/CONUS – green triangles, NE Pac – dark yellow triangles, NW Pac – purple triangles

- It is not clear why results from MERRA and ERA-Interim are shown in Figs. 9 and 10. You might want to omit Figure 9 to be more consistent, i.e. only ERA-Interim is shown in Figure 7.

We have deleted Figure 9 from the manuscript.

**Northern Hemisphere Contrail Properties Derived from Terra and Aqua MODIS Data for 2006 and 2012**

David P. Duda1, Sarah T. Bedka1, Patrick Minnis1, Douglas Spangenberg1, Konstantin Khlopenkov1, Thad Chee1, and William L. Smith, Jr.2

1Science Systems and Applications, Inc., Hampton, VA 23666-5986, USA 2NASA Langley Research Center, Hampton, VA 23681-2199, USA *Correspondence to*: David P. Duda (david.p.duda@nasa.gov)

[revised manuscript text omitted]

**2.3 Contrail radiative forcing**

15 The contrail radiative forcing calculations used the same procedure as with the 2006 data study (Spangenberg et al., 2013). For each contrail pixel that was assumed to be completely covered by a contrail, the radiative forcing is defined to be  $CRF = F_{conf} - F_{con}, \qquad (1)$

where  $F_{conf}$  and  $F_{con}$  are the upward top-of-atmosphere shortwave or longwave fluxes for contrail-free and contrailcovered conditions, respectively. The fluxes are computed using the Fu-Liou RTM (Fu and Liou, 1993; Fu et al., 1998).

- 20  $F_{con}$  is derived assuming an atmosphere with contrail-covered conditions where a contrail layer and background cloud, if applicable, are inserted at the relevant altitudes.  $F_{conf}$  is computed for the contrail-free situation for the same background conditions but without the contrail layer. Both the solar and longwave CRF are computed using equation (1), as well as the net CRF, which is the sum of solar and longwave CRF. The contrail properties,  $\tau$ , De, and contrail temperature, define the contrail layer for each pixel calculation. Further details of the CRF calculation are discussed in Spangenberg et al. (2013).
- 25 An updated version of the Fu-Liou radiative transfer program was employed, but it is not expected to affect the computed radiative forcing significantly. The most important change to the CRF assessment would be due to differences in the determination of the background cloud properties discussed above. No CRF calculations were possible for the postprocessed contrail cirrus Masks D and E due to difficulties in determining cloudiness background in situations where contrail cirrus extended over a large area.

**3 Results**

**3.1 Contrail mask**

The first and most basic parameter determined from this study is CC. The CC mask determines the amount and location of linear contrails and provides the foundation for the subsequent contrail property and radiative forcing retrievals. 5 The results of CC Mask B for 2012 are summarized in Figure 1.

Some consistent differences appear between the *Terra* (with overpasses at approximately 10:30 and 22:30 local time) and *Aqua* (with overpasses at approximately 01:30 and 13:30 local time) coverage in 2012. For example, *Terra* coverage is greater than *Aqua* over most air traffic regions including CONUS, Europe, China, and the eastern half of the air route between Hawaii and western CONUS. *Aqua* coverage is greater than *Terra* over the central North Atlantic, portions of the Europe to Latin America (LA) air route, northern Asia, and northern Africa.

Assuming that on average the upper tropospheric temperature and humidity do not change significantly during the 3 hours between the two overpass times, then the differences found in the *Terra* and *Aqua* CC are mostly likely due to differences in air traffic density relative to each satellite's overpass time. Figure 2 shows the difference in the 2012 annual mean air traffic density occurring 1, 2, 3, and 4-h before each satellite's (*Terra* minus *Aqua*) overpass time. The best match

- 15 between the patterns in Figure 2 and Figure 1c appears to vary from 1 to 3 h, depending on location. Overall, good matches occur for the case where the air traffic densities are compared 2 h before the overpass times, suggesting that most contrails are about 2 h old when detected by the satellite CDA. Previous studies (Duda et al., 2004, Vázquez-Navarro et al., 2015) however, have reported a contrail mean age of 1 h in contrails identified in geostationary satellite data, indicating that many mid-latitude contrails are detectable as early as 1 h after formation.
- 20

10

By comparing the results of this study with the 2006 data, we can examine the change in linear CC between the two years. The 2012, *Terra* Northern Hemisphere CC (Mask B) shows a 3% relative increase compared to 2006 data, from 0.136 / percent to 0.140 percent, while the 2012, *Aqua* coverage increased by almost 7 percent, from 0.134 percent to 0.143 percent.

Figure 3 shows the changes in annual mean NH CC between 2006 and 2012 for both satellites. The satellites show similar changes in the magnitude and distribution of the CC between 2006 and 2012. Both the *Jerra* and *Aqua* plots show

25 larger increases in CC along the North Atlantic corridor and parts of the Indian Ocean, with smaller increases over northwestern CONUS, northwestern Asia, and tropical Africa. Decreases in CC from 2006 to 2012 are apparent over southern CONUS, Western Europe, and northeastern Canada,

To examine the two-year differences in CC more closely, the NH was divided into nine air traffic regions to determine where CC changes were most pronounced (Figure 4). Table 1 summarizes the relative changes ([2012 – 2006]/2006×100%) in the CC (day+night) for each of the nine regions. In addition, the relative changes in unscreened

contrail coverage are presented in the table to highlight the impact of the air traffic screening on the determination of CC. The most prominent year-to-year differences between the CC and the unscreened contrail coverage are in the transoceanic air traffic regions (HI/CONUS, Europe/LA) and western Asia. These two transoceanic regions have significant declines in CC while the corresponding unscreened coverage changes are smaller. In contrast, western Asia shows moderate Formatted: Font: Times New Roman

| λ             | Deleted: screened                                                                                                                                                                                                                                                                                                                                                                                                                                                                    |
|---------------|--------------------------------------------------------------------------------------------------------------------------------------------------------------------------------------------------------------------------------------------------------------------------------------------------------------------------------------------------------------------------------------------------------------------------------------------------------------------------------------|
| X             | Deleted: .                                                                                                                                                                                                                                                                                                                                                                                                                                                                           |
| λ             | Deleted: Terra minus Aqua                                                                                                                                                                                                                                                                                                                                                                                                                                                            |
| λ             | Deleted: difference estimated at                                                                                                                                                                                                                                                                                                                                                                                                                                                     |
| λ             | Deleted: the                                                                                                                                                                                                                                                                                                                                                                                                                                                                         |
| -(            | Deleted: times                                                                                                                                                                                                                                                                                                                                                                                                                                                                       |
| -(            | Deleted: 1                                                                                                                                                                                                                                                                                                                                                                                                                                                                           |
| -(            | Deleted: 2012 annual-mean                                                                                                                                                                                                                                                                                                                                                                                                                                                            |
|               | Deleted: screened                                                                                                                                                                                                                                                                                                                                                                                                                                                                    |
| /(            | Deleted: screened                                                                                                                                                                                                                                                                                                                                                                                                                                                                    |
| /(            | Deleted: An examination of                                                                                                                                                                                                                                                                                                                                                                                                                                                           |
| /(            | Deleted: screened                                                                                                                                                                                                                                                                                                                                                                                                                                                                    |
|               | Deleted: (Figure 3) versus the changes in unscreened CC (Figure 4) provides some insight into how air traffic density affected the screened coverage estimations. In comparison with the screened CC, the hemispheric-mean unscreened CC changed only slightly between 2006 and 2012. Terra NH CC decreased 2 percent compared to 2006 data from 0.337% to 0.329%, and the 2012 unscreened Aqua CC increased 1 percent, from 0.312% to 0.316%. Bothfor both satellites |
| (             | Deleted: screened and unscreened                                                                                                                                                                                                                                                                                                                                                                                                                                                     |
| ~(            | Deleted: screened and unscreened CC in Figures 3 and 4                                                                                                                                                                                                                                                                                                                                                                                                                               |
| -(            | Deleted: screened and unscreened                                                                                                                                                                                                                                                                                                                                                                                                                                                     |
|               | Deleted: The most notable differences between Figures 3 and 4 occur in the air routes between Europe and Latin America, and between CONUS and Hawaii, where screened CC decreases between 2006 and 2012, while the unscreened CC changes are mixed. 9                                                                                                                                                                                                              |
| $\langle$     | Deleted: interannual                                                                                                                                                                                                                                                                                                                                                                                                                                                                 |
| (             | Deleted: 5                                                                                                                                                                                                                                                                                                                                                                                                                                                                           |
| (             | Deleted: screened and unscreened                                                                                                                                                                                                                                                                                                                                                                                                                                                     |
| $\mathcal{X}$ | Deleted: of the Northern Hemisphere                                                                                                                                                                                                                                                                                                                                                                                                                                                  |
| -(            | Deleted: screened                                                                                                                                                                                                                                                                                                                                                                                                                                                                    |
| Y             | Deleted: interannual CC                                                                                                                                                                                                                                                                                                                                                                                                                                                              |
| (             | Deleted: screened coverage                                                                                                                                                                                                                                                                                                                                                                                                                                                           |

declines in unscreened coverage but small increases in CC. To confirm that the differences between the CC and the unscreened coverage trends in these air traffic regions can be explained by changes in the air traffic density between 2006 and 2012, Figure  $\leq$  shows the difference in annual-mean air traffic density between 2012 and 2006 for each satellite 2 h before their respective overpass times. As presented above, an examination of the differences in the *Terra* and *Aqua*, CC for

5 2012 suggests that most detected contrails form approximately 2 h before satellite overpass time. Figure 5 shows that air traffic density increased over nearly all of the Northern Hemisphere between 2006 and 2012 except for the air corridor between Europe and Latin America, and over parts of the HI/CONUS corridor.

Table 2 provides a list of the mean air traffic changes between 2006 and 2012 based on a sample of waypoint data from the nine air traffic regions. Seven of the nine air traffic regions show increases in air traffic. The largest increases occur

- 10 over E Asia and W Asia. Two regions show a decrease in air traffic: a small decrease in the Hawaii to CONUS corridor, and a nearly 60 percent decrease in the Europe to LA corridor. Thus, the decrease in air traffic in these two regions results in diminished CC but smaller changes in unscreened contrail coverage. [The large increase in air traffic over western Asia may also explain in part why the CC remained relatively unchanged in 2012 despite the observed moderate decrease in unscreened coverage.]
- 15 Like changes in air traffic, year-to-year changes in the upper tropospheric thermodynamic state also affect the detected linear CC. To consider this possible factor, we present in Figure 6 and Table 2 the 2012 minus 2006 change in potential persistent contrail frequency (PPCF) between 200 and 250 hPa, the tropospheric layer where most of the contrail-forming air traffic occurs. The PPCF is computed using temperature and relative humidity statistics from ERA-Interim (ECMWF) reanalysis data (Dee et al., 2011), and is an indicator of how often conditions that are favorable for the 20 development of persistent contrails occur. It is assumed that MERRA relative humidities are generally consistent with their
- ERA-Interim counterparts.

The relative differences in PPCF between 2006 and 2012 are listed in Table 2. In many of the air traffic regions, the relative differences in PPCF between 2006 and 2012 are small, suggesting that the differences between the 2006 and 2012 are small, suggesting that the differences between the 2006 and 2012 are small, suggesting that the differences between the 2006 and 2012 are small, suggesting that the differences between the 2006 and 2012 are small, suggesting that the differences between the 2006 and 2012 are small, suggesting that the differences between the 2006 and 2012 are small, suggesting that the differences between the 2006 and 2012 are small, suggesting that the differences between the 2006 and 2012 are small, suggesting that the differences between the 2006 and 2012 are small, suggesting that the differences between the 2006 and 2012 are small, suggesting that the differences between the 2006 and 2012 are small, suggesting that the differences between the 2006 and 2012 are small, suggesting that the differences between the 2006 and 2012 are small, suggesting that the differences between the 2006 and 2012 are small, suggesting that the differences between the 2006 and 2012 are small, suggesting that the differences between the 2006 and 2012 are small, suggesting that the differences between the 2006 and 2012 are small.

25 troposphere. However, for some regions, the year-to-year changes in PPCF are more significant, and may have played a larger role in the difference in number of detected contrails between 2006 and 2012. The PPCF changes (Figure 3) correlate with CC changes (Figure 3) over northwestern and central Asia, the Indian Ocean, southern CONUS, northern Europe and off the coast of Western Europe, Greenland, and parts of northern Canada. For the air traffic regions, the increase in CC over the North Atlantic is correlated with an increase in PPCF between 2006 and 2012, but in the Europe to LA air corridor the increase in PPCF in 2012 appears to have minimal impact on the decrease in CC.

For more insight into the relationship between the detected CC and PPCF, both the CC and PPCF data were sorted by air traffic region and season [DJF, MAM, JJA, SON] for both 2006 and 2012 and compared. As a test of how consistent the detection of CC by the CDA was between both satellites for both years, we first computed the two-year relative change ([2012 – 2006]/2006x100%) in seasonal CC derived from *Terra* MODIS data versus the corresponding change in contrail.

35 coverage computed from Aqua MODIS data for each of the high air traffic regions. Figure 7a shows a scatter plot of the

| Deleted: | screened coverage.    |
|----------|-----------------------|
| Deleted: | : investigate whether |
| Deleted: | : screened            |
| Deleted: | : CC                  |
| Deleted: | : 6                   |
| Deleted: | at )                  |
| Deleted: | : Terra and Aqua      |
| Deleted: | : time                |
| Deleted: | screened              |
| Deleted: | : 6                   |
| -        |                       |

| (      | Deleted: screened          |
|--------|----------------------------|
| ~~~(   | Deleted: CC.               |
| ······ | Deleted: screened coverage |

| ( | Deleted: | 7 |
|---|----------|---|
|   |          |   |

**Deleted: screened Deleted: 7 Deleted: unscreened Deleted: 4 Deleted: screened and unscreened coverage Deleted: screened Formatted: Font: Times New Roman Formatted: Font: Times New Roman Formatted: Font: Times New Roman Deleted: screened and unscreened contrail coverage Formatted: Font: Times New Roman Deleted: screened and unscreened contrail coverage Formatted: Font: Times New Roman Deleted: sa Formatted: Font: Times New Roman**

relative difference in seasonal Terra CC between 2012 and 2006 data for each region versus the corresponding two-year difference in Aqua-derived CC. Figure 7b shows the same scatter plot with the individual linear regressions for each of the air traffic regions. The year-to-year CC changes from both satellites are well correlated with each other.

Similar scatter plots of the two-year relative change in CC from *Terra* compared to the corresponding seasonal two year absolute (2012 - 2006) change in PPCF computed from ERA-Interim data are shown in Figure 8. Figure 8a shows a scatter plot of the year-to-year change in *Terra* CC versus the ECMWF-derived PPCF for each season and each air traffic region. Figure 8b shows the same scatter plot with the individual linear regressions for each of the air traffic regions. The removal of the outlier regression in the plot (the red triangles representing the Europe/Latin America corridor) would

increase the correlation coefficient of the remaining data to 0.663. The differences between the Europe/Latin America data and the other air traffic regions suggest that the nature of the air traffic data between 2006 and 2012 may have changed for

- the Europe/Latin America air routes, or that the contrails in this region became harder to detect in 2012. The shift of the North Atlantic regression to the right of the other regions suggests that contrails might have been more easily detected in the North Atlantic during 2012 compared to 2006. The standard deviation of the background 12-um brightness temperature, which is known to affect the detectability of linear contrails by the CDA (Mannstein et al., 1999), decreased by about 10% in
- 15 2012 compared to 2006 in the North Atlantic region, which may account for some of the discrepancy. In addition, the magnitude of the discrepancy between the North Atlantic and the other air traffic regions is noticeably larger in the MERRA-based plots (not shown) than in Figure 8, It appears that there is greater uncertainty between the MERRA- and ECMWF-derived PPCF in this region between 2006 and 2012 than in other regions.

The results presented here demonstrate that interannual changes in air traffic density and PPCF appear to have some 20 influence on the change in satellite-detected CC between 2006 and 2012, although some of the changes between the two years are more difficult to explain, especially the large increase in CC over the North Atlantic air corridor. Part of the increase in CC between 2006 and 2012 over the North Atlantic may be due to changes in flight altitudes in 2012 that shifted more flights into levels of the atmosphere where ambient conditions are more likely for persistent contrail formation than in 2006. Overall, the uncertainty in the CC estimates are probably large enough that the differences between 2006 and 2012 are 25 not likely to be statistically significant, in large part because of the difficulty in evaluating the air traffic data that are critical

25 not likely to be statistically significant, in large part because of the difficulty in evaluating the air traffic data that are critical to the screening process.

**3.2 Contrail cirrus coverage estimation**

30

In addition to linear contrails, CC due to contrail cirrus would increase the overall contrail radiative forcing (CRF) as contrails spread into non-linear, overlapping cloudiness that cannot be detected by the CDA. To estimate contrail cirrus effects, Minnis et al. (2013) tracked contrails over the United States in geostationary satellite imagery and determined the properties of contrail cirrus over selected areas; visual inspection was used to ensure that only contrails produced the existing cirrus clouds in each region. Overall for 21 cases, the combined linear and contrail cirrus coverage was on average 3.5 times the value determined from mask B. The contrail cirrus  $\tau$  and De values were larger than the corresponding linear contrail values.

[revised manuscript text omitted]

- 15 description of our satellite-based contrail observation system, no comparison with model estimates of contrail coverage was included. More research is needed to compare satellite observations of contrails with such modelling estimates, but such a study would require much additional work to determine how satellite-detectable contrail coverage relates to estimates of potential coverage from meteorological conditions and air traffic density. To make a fair comparison, the model results would have to be screened for natural cirrus and other high ice clouds that would render most contrails invisible in the
- 20 satellite imagery. In addition, model-based estimates would have to consider the overall detectability limitations of the imagery and the temporal and spatial sampling of the satellite observations.

The algorithm developed to detect contrail-like cirrus in this study is a preliminary attempt to define contrail cirrus. Although useful as a heuristic tool to examine how contrail cirrus detection varies between different times and locations, it requires refinement, especially over different surface backgrounds and varying viewing angles. Furthermore, the study of

- 25 contrail cirrus development could be aided by the launch of next generation imagers onboard the Himawari-8 and GOES-16 satellites. These platforms can provide full disk 10 and 15-minute loops, respectively, of high-resolution multi-spectral geosynchronous imagery that would allow detailed analysis of contrail spreading by identifying individual contrails with specific flights from the waypoint database. Expanding on the analysis presented in Minnis et al. (2013), the lifecycles of a large set of identified contrails could be related to several meteorological variables (such as RHI, vertical wind, wind shear,
- 30 depth of the super-saturated layer) to determine which factors influence the growth and spreading of persistent linear contrails into contrail cirrus. Such a dataset would help us improve our understanding of how contrail cirrus contributes to the observed increase in global cirrus coverage and would provide valuable data to contrail models that explicitly simulate the lifespan of contrails, thus advancing our knowledge of the impacts of contrail cirrus on climate.
  - 13

**Supplement**

The supplement related to this article is available online at: https://doi.org/10.5194/acp-2018-993-supplement.

**Author contribution**

5 David Duda prepared the manuscript with contributions from co-authors. David Duda, Sarah Bedka, and Doug Spangenberg performed the analysis of the contrail masks, contrail property retrievals, and contrail radiative forcing, respectively. Patrick Minnis conceptualized the overall research goals and aims of the study, supervised the research project and acquired funding for the study. Konstantin Khlopenkov developed programming code for analysing the commercial aircraft waypoint data and advecting contrail tracks via the MERRA wind data. Thad Chee managed the implementation of 10 the computer code and supporting algorithms to process the satellite data and to create the contrail masks. William Smith also supervised the research project and provided review and commentary of the manuscript.

**Competing interests**

The authors declare that they have no conflict of interest.

**Acknowledgements**

15 The waypoint data used for this work were provided by U.S. DOT Volpe Center and are based on data provided by the U.S. FAA and EUROCONTROL in support of the objectives of the International Civil Aviation Organization (ICAO) Committee on Aviation Environmental Protection CO2 Task Group. Any opinions, findings, and conclusions or recommendations expressed in this material are those of the authors and do not necessarily reflect the views of the U.S. DOT Volpe Center, the U.S. FAA, EUROCONTROL, or ICAO. The authors also thank three anonymous reviewers for their assistance in evaluating this paper.

[revised manuscript text omitted]